# Permutation Equivariant Neural Functionals

**Allan Zhou**[1]    **Kaien Yang**[1]    **Kaylee Burns**[1]    **Adriano Cardace**[2]    **Yiding Jiang**[3]
**Samuel Sokota**[3]    **J. Zico Kolter**[3]    **Chelsea Finn**[1]
[1]Stanford University    [2]University of Bologna    [3]Carnegie Mellon University
ayz@cs.stanford.edu

## Abstract

This work studies the design of neural networks that can process the weights or gradients of other neural networks, which we refer to as *neural functional networks* (NFNs). Despite a wide range of potential applications, including learned optimization, processing implicit neural representations, network editing, and policy evaluation, there are few unifying principles for designing effective architectures that process the weights of other networks. We approach the design of neural functionals through the lens of symmetry, in particular by focusing on the permutation symmetries that arise in the weights of deep feedforward networks because hidden layer neurons have no inherent order. We introduce a framework for building *permutation equivariant* neural functionals, whose architectures encode these symmetries as an inductive bias. The key building blocks of this framework are *NF-Layers* (neural functional layers) that we constrain to be permutation equivariant through an appropriate parameter sharing scheme. In our experiments, we find that permutation equivariant neural functionals are effective on a diverse set of tasks that require processing the weights of MLPs and CNNs, such as predicting classifier generalization, producing "winning ticket" sparsity masks for initializations, and classifying or editing implicit neural representations (INRs). In addition, we provide code for our models and experiments[1].

## 1   Introduction

As deep neural networks have become increasingly prevalent across various domains, there has been a growing interest in techniques for processing their weights and gradients as data. Example applications include learnable optimizers for neural network training [3, 56, 2, 44], extracting information from implicit neural representations of data [61, 45, 58], corrective editing of network weights [57, 11, 46], policy evaluation [24], and Bayesian inference given networks as evidence [60]. We refer to functions of a neural network's weight-space (such as weights, gradients, or sparsity masks) as *neural functionals*; when these functions are themselves neural networks, we call them *neural functional networks* (NFNs).

In this work, we design neural functional networks by incorporating relevant symmetries directly into the architecture, following a general line of work in "geometric deep learning" [8, 54, 34, 5]. For neural functionals, the symmetries of interest are transformations of a network's weights that preserve the network's behavior. In particular, we focus on *neuron permutation symmetries*, which are those that arise from the fact that the neurons of hidden layers have no inherent order.

Neuron permutation symmetries are simplest in feedforward networks, such as multilayer perceptrons (MLPs) and basic convolutional neural networks (CNNs). These symmetries are induced by the fact that the neurons in each hidden layer of a feedforward network can be arbitrarily permuted without changing its behavior [27]. In MLPs, permuting the neurons in hidden layer $i$ corresponds to

---

[1] https://github.com/AllanYangZhou/nfn

37th Conference on Neural Information Processing Systems (NeurIPS 2023).

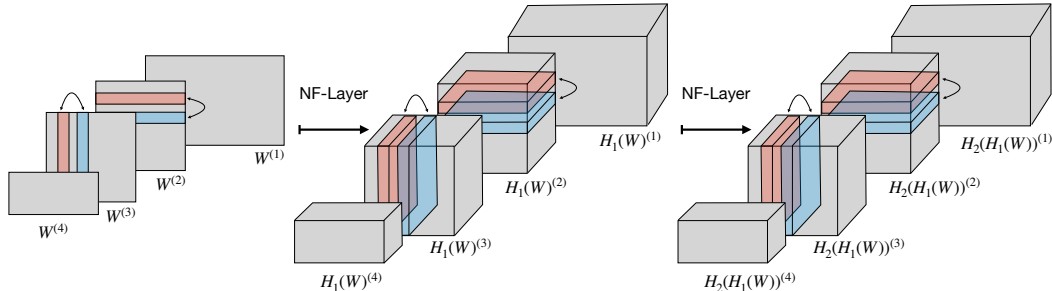

Figure 1: The internal operation of our permutation equivariant neural functionals (NFNs). The NFN processes the input weights through a series of equivariant NF-Layers, with each one producing *weight-space features* with varying numbers of channels. In this example, a neuron permutation symmetry simultaneously permutes the rows of $W^{(2)}$ and the columns of $W^{(3)}$. This permutation propagates through the NFN in an equivariant manner.

permuting the rows of the weight matrix $W^{(i)}$, and the columns of the next weight matrix $W^{(i+1)}$ as shown on the left-hand side of Figure 1. Note that the same permutation must be applied to the rows $W^{(i)}$ and columns of $W^{(i+1)}$, since applying *different* permutations generally changes network behavior and hence does not constitute a neuron permutation symmetry.

We introduce a new framework for constructing neural functional networks that are invariant or equivariant to neuron permutation symmetries. Our framework extends a long line of work on permutation equivariant architectures [52, 68, 25, 63, 41] that design equivariant layers for a particular permutation symmetry of interest. Specifically, we introduce neural functional layers (NF-Layers) that operate on weight-space features (see Figure 1) while being equivariant to neuron permutation symmetries. Composing these NF-Layers with pointwise non-linearities produces equivariant neural functionals.

We propose different NF-Layers depending on the assumed symmetries of the input weight-space: either only the hidden neurons of the feedforward network can be permuted (hidden neuron permutation, HNP), or all neurons, including inputs and outputs, can be permuted (neuron permutation, NP). Although the HNP assumption is typically more appropriate, the corresponding NF-Layers can be parameter inefficient and computationally infeasible in some settings. In contrast, NF-Layers derived under NP assumptions often lead to much more efficient architectures, and, when combined with a positional encoding scheme we design, can even be effective on tasks that require breaking input and output symmetry. For situations where invariance is required, we also define invariant NF-Layers that can be applied on top of equivariant weight-space features.

Finally, we investigate the applications of permutation equivariant neural functionals on tasks involving both feedforward MLPs and CNNs. Our first two tasks require (1) predicting the test accuracy of CNN image classifiers and (2) classifying implicit neural representations (INRs) of images and 3D shapes. We then evaluate NFNs on their ability to (3) predict good sparsity masks for initializations (also called *winning tickets* [19]), and on (4) a weight-space "style-editing" task where the goal is to modify the content an INR encodes by directly editing its weights. In multiple experiments across these diverse settings, we find that permutation equivariant neural functionals consistently outperform non-equivariant methods and are effective for solving weight-space tasks.

**Relation to DWSNets.** The recent work of Navon et al. [45] recognized the potential of leveraging weight-space symmetries to build equivariant architectures on deep weight-spaces; they characterize a weight-space layer which is mathematically equivalent to our NF-Layer in the HNP setting. Their work additionally studies interesting universality properties of the resulting equivariant architectures, and demonstrates strong empirical results for a suite of tasks that require processing the weights of MLPs. Our framework additionally introduces the NP setting, where we make stronger symmetry assumptions to develop equivariant layers with improved parameter efficiency and practical scalability. We also extend our NFN variants to process convolutional neural networks (CNNs) as input, leading to applications such as predicting the generalization of CNN classifiers (Section 3.1).

Table 1: Permutation symmetries of $L$-layer feedforward networks with $n_0, \ldots, n_L$ neurons at each layer. All feedforward networks are invariant under hidden neuron permutations (HNP), while NP assumes that input and output neurons can also be permuted. We show the corresponding equivariant NF-Layers which process weight-space features from $\mathcal{U}$, with $c_i$ input channels and $c_o$ output channels.

| Group | Abbrv | Permutable layers | Equivariant NF-Layer | |
|---|---|---|---|---|
| | | | Signature | Parameter count |
| $\mathcal{S} = \prod_{i=0}^{L} S_{n_i}$ | NP | All layers | $H : \mathcal{U}^{c_i} \to \mathcal{U}^{c_o}$ | $O(c_i c_o L^2)$ |
| $\tilde{\mathcal{S}} = \prod_{i=1}^{L-1} S_{n_i}$ | HNP | Hidden layers | $\tilde{H} : \mathcal{U}^{c_i} \to \mathcal{U}^{c_o}$ | $O\left(c_i c_o (L + n_0 + n_L)^2\right)$ |
| — | — | None | $T : \mathcal{U}^{c_i} \to \mathcal{U}^{c_o}$ | $c_i c_o \dim(\mathcal{U})^2$ |

## 2 Equivariant neural functionals

We begin by setting up basic concepts related to (hidden) neuron permutation symmetries, before defining the equivariant NF-Layers in Sec. 2.2 and invariant NF-Layers in Sec. 2.3.

### 2.1 Preliminaries

Consider an $L$-layer feedforward network having $n_i$ neurons at layer $i$, with $n_0$ and $n_L$ being the input and output dimensions, respectively. The network is parameterized by weights $W = \left\{ W^{(i)} \in \mathbb{R}^{n_i \times n_{i-1}} \mid i \in [\![1..L]\!] \right\}$ and biases $v = \left\{ v^{(i)} \in \mathbb{R}^{n_i} \mid i \in [\![1..L]\!] \right\}$. We denote the combined collection $U := (W, v)$ belonging to weight-space, $\mathcal{U} := \mathcal{W} \times \mathcal{V}$.

Since the neurons in a hidden layer $i \in \{1, \cdots, L-1\}$ have no inherent ordering, the network is invariant to the symmetric group $S_{n_i}$ of permutations of the neurons in layer $i$. This reasoning applies to every hidden layer, so the network is invariant to $\tilde{\mathcal{S}} := S_{n_1} \times \cdots \times S_{n_{L-1}}$, which we refer to as the **hidden neuron permutation** (HNP) group. Under the stronger assumption that the input and output neurons are also unordered, the network is invariant to $\mathcal{S} := S_0 \times \cdots \times S_{n_L}$, which we refer to as the **neuron permutation** (NP) group. We focus on the NP setting throughout the main text, and treat the HNP case in Appendix B. See Table 1 for a concise summary of the relevant notation for each symmetry group we consider.

Consider an MLP and a permutation $\sigma = (\sigma_0, \cdots, \sigma_L) \in \mathcal{S}$. The action of the neuron permutation group is to permute the rows of each weight matrix $W^{(i)}$ by $\sigma_i$, and the columns by $\sigma_{i-1}$. Each bias vector $v^{(i)}$ is also permuted by $\sigma_i$. So the action is $\sigma U := (\sigma W, \sigma v)$, where:

$$[\sigma W]^i_{jk} = W^{(i)}_{\sigma_i^{-1}(j), \sigma_{i-1}^{-1}(k)}, \quad [\sigma v]^i_j = v^{(i)}_{\sigma_i^{-1}(j)}. \tag{1}$$

Until now we have used $U = (W, v)$ to denote actual weights and biases, but the inputs to a neural functional layer could be any weight-space *feature* such as a gradient, sparsity mask, or the output of a previous NF-Layer (Figure 1). Moreover, we may consider inputs with $c \geq 1$ feature channels, belonging to $\mathcal{U}^c = \bigoplus_{i=1}^c \mathcal{U}$, the direct sum of $c$ copies of $\mathcal{U}$. Concretely, each $U \in \mathcal{U}^c$ consists of weights $W = \left\{ W^{(i)} \in \mathbb{R}^{n_i \times n_{i-1} \times c} \mid i \in [\![1..L]\!] \right\}$ and biases $v = \left\{ v^{(i)} \in \mathbb{R}^{n_i \times c} \mid i \in [\![1..L]\!] \right\}$, with the channels in the final dimension. The action defined in Eq. 1 extends to the multiple channel case if we define $W^{(i)}_{jk} := W^{(i)}_{j,k,:} \in \mathbb{R}^c$ and $v^{(i)}_j := v^{(i)}_{j,:} \in \mathbb{R}^c$.

The focus of this work is on making neural functionals that are equivariant (or invariant) to neuron permutation symmetries. Letting $c_i$ and $c_o$ be the number of input and output channels, we refer to a function $f : \mathcal{U}^{c_i} \to \mathcal{U}^{c_o}$ as $\mathcal{S}$-**equivariant** if $\sigma f(U) = f(\sigma U)$ for all $\sigma \in \mathcal{S}$ and $U \in \mathcal{U}^{c_i}$, where the action of $\mathcal{S}$ on the input and output spaces is defined by Eq. 1. Similarly, a function $f : \mathcal{U}^c \to \mathbb{R}$ is $\mathcal{S}$-**invariant** if $f(\sigma U) = f(U)$ for all $\sigma$ and $U$.

If $f, g$ are equivariant, then their composition $f \circ g$ is also equivariant; if $g$ is equivariant and $f$ is invariant, then $f \circ g$ is invariant. Since pointwise nonlinearities are already permutation equivariant, our remaining task is to design a *linear* NF-Layer that is $\mathcal{S}$-equivariant. We can then construct equivariant neural functionals by stacking these NF-Layers with pointwise nonlinearities.

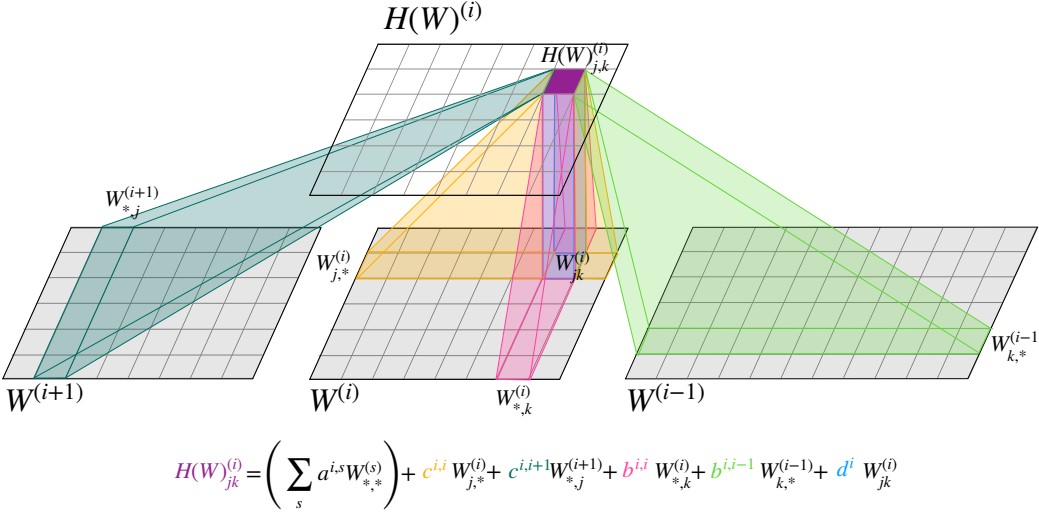

Figure 2: A permutation equivariant NF-Layer takes in weight-space features as input (bottom) and outputs transformed features (top), while respecting the neuron permutation symmetries of feedforward networks. This illustrates the computation of a single output element $H(W)_{jk}^i$, defined in Eq. 2. Each output is a weighted combination of rows or column sums of the input weights, which preserves permutation symmetry. The first term contributes a weighted combination of row-and-column sums from *every* input weight, though this is omitted for visual clarity.

## 2.2 Equivariant NF-Layers

We now construct a linear $\mathcal{S}$-equivariant layer that serves as a key building block for neural functional networks. In the single channel case, we begin with generic linear layers $T(\cdot;\theta) : \text{vec}(U) \mapsto \theta\text{vec}(U)$, where $\text{vec}(U) \in \mathbb{R}^{\dim(U)}$ is $U$ flattened as a vector and $\theta \in \mathbb{R}^{\dim(\mathcal{U}) \times \dim(\mathcal{U})}$ is a matrix of parameters. We show in Appendix B.2 that **any** $\mathcal{S}$-equivariant $T(\cdot;\theta)$ must satisfy a system of constraints on $\theta$ known as equivariant *parameter sharing*. We derive this parameter sharing by partitioning the entries of $\theta$ by the orbits of their indices under the action of $\mathcal{S}$, with parameters shared in each orbit [54]. Table 7 of the appendix describes the parameter sharing in detail.

Equivariant parameter sharing reduces the matrix-vector product $\theta\text{vec}(U)$ to the NF-Layer we now present. For simplicity we ignore $\mathcal{V}$ and assume here that $\mathcal{U} = \mathcal{W}$ and defer the full form to Eq. 3 in the appendix. Then $H : \mathcal{W}^{c_i} \to \mathcal{W}^{c_o}$ maps input $\left(W^{(1)}, \cdots, W^{(L)}\right)$ to $\left(H(W)^{(1)}, \cdots, H(W)^{(L)}\right)$. Recall that the inputs are not necessarily weights, but could be arbitrary weight-space features including the output of a previous NF-Layer. For $W^{(i)} \in \mathbb{R}^{n_i \times n_{i-1} \times c_i}$, the corresponding output is $H(W)^{(i)} \in \mathbb{R}^{n_i \times n_{i-1} \times c_o}$ with entries computed:

$$H(W)_{jk}^{(i)} = \left(\sum_s a^{i,s} W_{\star,\star}^{(s)}\right) + b^{i,i} W_{\star,k}^{(i)} + b^{i,i-1} W_{k,\star}^{(i-1)} + c^{i,i} W_{j,\star}^{(i)} + c^{i,i+1} W_{\star,j}^{(i+1)} + d^i W_{jk}^{(i)}. \quad (2)$$

Note that the terms involving $W^{(i-1)}$ or $W^{(i+1)}$ should be omitted for $i = 0$ and $i = L$, respectively, and $\star$ denotes summation or averaging over either the rows or columns. Recall that in the multi-channel case, each $W_{jk}^{(i)}$ is a vector in $\mathbb{R}^{c_i}$ so each parameter is a $c_o \times c_i$ matrix. We also provide a concrete pseudocode description of $H$ in Appendix A. Figure 2 visually illustrates the NF-Layer in the single-channel case, showing how the row or column sums from each input contribute to each output. To gain intuition for the operation of $H$, it is straightforward to check $\mathcal{S}$-equivariance:

**Proposition 1.** *The NF-Layer $H : \mathcal{U}^{c_i} \to \mathcal{U}^{c_o}$ (Eq. 2 and Eq. 3) is $\mathcal{S}$-equivariant, where the group's action on input and output spaces is defined by Eq. 1. Moreover, any linear $\mathcal{S}$-equivariant map $T : \mathcal{U}^{c_i} \to \mathcal{U}^{c_o}$ is equivalent to $H$ for some choice of parameters $a, b, c, d$.*

*Proof (sketch).* We can verify that $H$ satisfies the equivariance condition $[\sigma H(W)]_{jk}^{(i)} = H(\sigma W)_{jk}^{(i)}$ for any $i, j, k$ by expanding each side of the equation using the definitions of the layer and action

(Eq. 1). Moreover, Appendix B.2 shows that any $\mathcal{S}$-equivariant linear map $T(\cdot, \theta)$ must have the same equivariant parameter sharing as $H$, meaning that it must be equivalent to $H$ for some choice of parameter $a, b, c, d$. $\qquad\qquad\qquad\qquad\qquad\qquad\qquad\qquad\qquad\qquad\qquad\qquad\qquad\quad$ $\square$

Informally, the above proposition tells us that $H$ can express any linear $\mathcal{S}$-equivariant function of a weight-space. Since $\tilde{\mathcal{S}}$ is a subgroup of $\mathcal{S}$, $H$ is also $\tilde{\mathcal{S}}$-equivariant. However, it does not express every possible linear $\tilde{\mathcal{S}}$-equivariant function. We derive the full $\tilde{\mathcal{S}}$-equivariant NF-Layer $\tilde{H} : \mathcal{U} \to \mathcal{U}$ in Appendix C.

Table 1 summarizes the number of parameters (after parameter sharing) under different symmetry assumptions. While in general a linear layer $T(\cdot; \theta) : \mathcal{U}^{c_i} \to \mathcal{U}^{c_o}$ has $c_i c_o \dim(\mathcal{U})^2$ parameters, the equivariant NF-Layers have significantly fewer free parameters due to parameter sharing. The $\mathcal{S}$-equivariant layer $H$ has $O\left(c_i c_o L^2\right)$, while the $\tilde{\mathcal{S}}$-equivariant layer $\tilde{H}$ has $O\left(c_i c_o (L + n_0 + n_L)^2\right)$ parameters. The latter's quadratic dependence on input and output dimensions can be prohibitive in some settings, such as in classification where the number of outputs can be tens of thousands.

**Extension to convolutional weight-spaces.** In convolution layers, since neurons correspond to spatial *channels*, we let $n_i$ denote the number of channels at the $i^{\text{th}}$ layer. Each bias $v^{(i)} \in \mathbb{R}^{n_i}$ has the same dimensions as in the fully connected case, so only the convolution filter needs to be treated differently since it has additional spatial dimension(s) that cannot be permuted. For example, consider a 1D CNN with filters $W = \left\{ W^{(i)} \in \mathbb{R}^{n_i \times n_{i-1} \times w} \mid i \in [\![1..L]\!] \right\}$, where $n_i \times n_{i-1}$ are the output and input channel dimensions and $w$ is the filter width. We let $W^{(i)}_{jk} := W^{(i)}_{j,k,:} \in \mathbb{R}^w$ denote the $k^{\text{th}}$ filter in the $j^{\text{th}}$ output channel, then define the $\mathcal{S}$-action the same way as in Eq. 1.

We immediately observe the similarities to multi-channel features: both add dimensions that are not permuted by the group action. In fact, suppose we have $c$-channel features $U \in \mathcal{U}^c$ where $\mathcal{U}$ is the weight-space of a 1D CNN. Then we combine the filter and channel dimensions of the weights, with $W^{(i)} \in \mathbb{R}^{n_i \times n_{i-1} \times (cw)}$. This allows us to use the multi-channel NF-Layer $H : \mathcal{U}^{wc_i} \to \mathcal{U}^{wc_o}$. Any further channel dimensions, such as those for 2D convolutions, can also be folded into the channel dimension.

It is common for CNNs in image classification to follow convolutional layers with pooling and fully connected (FC) layers, which opens the question of defining the $\mathcal{S}$-action when layer $\ell$ is FC and layer $\ell - 1$ is convolutional. If global spatial pooling removes all spatial dimensions from the output of $\ell - 1$ (as in e.g., ResNets [26] and the Small CNN Zoo [64]), then we can verify that the existing action definitions work without modification. We leave more complicated situations (e.g., when nontrivial spatial dimensions are flattened as input to FC layers) to future work.

**IO-encoding.** The $\mathcal{S}$-equivariant layer $H$ is more parameter efficient than $\tilde{H}$ (Table 1), but its NP assumptions are typically too strong. To resolve this problem, we can add either learned or fixed (sinusoidal) position embeddings to the columns of $W^{(1)}$ and the rows of $W^{(L)}$ and $v^{(L)}$; this breaks the symmetry at input and output neurons even when using $\mathcal{S}$-equivariant layers. In our experiments, we find that IO-encoding makes $H$ competitive or superior to $\tilde{H}$, while using a fraction of the parameters.

## 2.3 Invariant NF-Layers

Invariant neural functionals can be designed by composing multiple equivariant NF-Layers with an invariant NF-Layer, which can then be followed by an MLP. We define an $\mathcal{S}$-invariant[2] layer $P : \mathcal{U} \to \mathbb{R}^{2L}$ by simply summing or averaging the weight matrices and bias vectors across any axis that has permutation symmetry, i.e., $P(U) = \left( W^{(1)}_{\star,\star}, \cdots, W^{(L)}_{\star,\star}, v^{(1)}_{\star}, \cdots, v^{(L)}_{\star} \right)$. We can then apply a fully connected layer to the output of $P$ to produce an invariant vector of arbitrary size. In fact, this combination expresses every possible linear invariant function on $\mathcal{U}$:

**Proposition 2.** *Any $\mathcal{S}$-invariant linear function $\mathcal{U} \to \mathbb{R}^d$ can be expressed in the form $f \circ P$, for some choice of linear $f : \mathbb{R}^{2L} \to \mathbb{R}^d$. Proof: See section B.3.*

---

[2] We define the analogous $\tilde{\mathcal{S}}$-invariant layer $\tilde{P}$ in Eq. 20 of the appendix.

Table 2: Test $\tau$ of generalization prediction methods on the Small CNN Zoo [64], which contains the weights and test accuracies of many small CNNs trained on different datasets, such as CIFAR-10-GS or SVHN-GS. NFN$_{\text{HNP}}$ outperforms other methods on both datasets. Uncertainties indicate max and min over two runs.

|  | NFN$_{\text{HNP}}$ | NFN$_{\text{NP}}$ | STATNN |
|---|---|---|---|
| CIFAR-10-GS | **0.934 $\pm$ 0.001** | 0.922 $\pm$ 0.001 | 0.915 $\pm$ 0.002 |
| SVHN-GS | **0.931 $\pm$ 0.005** | 0.856 $\pm$ 0.001 | 0.843 $\pm$ 0.000 |

## 3 Experiments

Our experiments evaluate permutation equivariant neural functionals on a variety of tasks that require either invariance (predicting CNN generalization and extracting information from INRs) or equivariance (predicting "winning ticket" sparsity masks and weight-space editing of INR content).

Throughout the experiments, we construct neural functional networks (NFNs) using the NF-Layers described in the previous section. Although the specific design varies depending on the task, we will broadly refer to our permutation equivariant NFNs as NFN$_{\text{NP}}$ and NFN$_{\text{HNP}}$, depending on which NF-Layer variant they use (see Table 1). We also evaluate a "pointwise" ablation of our equivariant NF-Layer that ignores interactions between weights by only using the last term of Eq. 2, computing $H(W)^i_{jk} := d^i W^{(i)}_{jk}$. We refer to NFNs that use this pointwise NF-Layer as NFN$_{\text{PT}}$.

Using our benchmarks, we compare the performance of NFNs with other methods for processing weights, including standard MLPs that operate on the flattened weight inputs. To encourage permutation equivariance, we optionally augment the MLP's training with permutations using Eq. 1. On relevant datasets we also compare with the recently developed DWSNets [47], an equivariant architecture similar to NFN$_{\text{HNP}}$, and with inr2vec [12], a recent non-equivariant approach for learning useful representations from weights.

### 3.1 Predicting CNN generalization from weights

Why deep neural networks generalize despite being heavily overparameterized is a longstanding research problem in deep learning. One recent line of work has investigated the possibility of directly predicting the test accuracy of the models from the weights [64, 16]. The goal is to study generalization in a data-driven fashion and ultimately identify useful patterns from the weights.

Prior methods develop various strategies for extracting potentially useful features from the weights before using them to predict the test accuracy [29, 67, 64, 30, 42]. However, using hand-crafted features could fail to capture intricate correlations between the weights and test accuracy. Instead, we explore using neural functionals to predict test accuracy from the *raw weights* of feedforward convolutional neural networks (CNN) from the *Small CNN Zoo* dataset [64], which contains thousands of CNN weights trained on several datasets with a shared architecture, but varied optimization hyperparameters. We compare the predictive power of NFN$_{\text{HNP}}$ and NFN$_{\text{NP}}$ against a method of Unterthiner et al. [64] that trains predictors on statistical features extracted from each weight and bias, and refer to it as STATNN. To measure the predictive performance of each method, we use *Kendall's $\tau$* [31], a popular rank correlation metric with values in $[-1, 1]$.

In Table 2, we show the results on two challenging subsets of Small CNN Zoo corresponding to CNNs trained on CIFAR-10-GS and SVHN-GS (GS stands for grayscaled). We see that NFN$_{\text{HNP}}$ consistently performs the best on both datasets by a significant margin, showing that having access to the full weights can increase predictive power over hand-designed features as in STATNN. Because the input and output dimensionalities are small on these datasets, NFN$_{\text{HNP}}$ only uses moderately more ($\sim 1.4\times$) parameters than NFN$_{\text{NP}}$ with equivalent depth and channel dimensions, while having significantly better performance.

### 3.2 Classifying implicit neural representations of images and 3D shapes

Given the rise of implicit neural representations (INRs) that encode data such as images and 3D-scenes [61, 43, 7, 49, 58, 45, 14, 15], it is natural to wonder how to extract information about the

Table 3: Classification test accuracies (%) for implicit neural representations of MNIST, FashionMNIST, and CIFAR-10. Equivariant architectures such as NFNs and DWSNets [47] outperform the non-equivariant MLP baselines, even when the MLP has permutation augmentations. Our $\text{NFN}_{\text{NP}}$ variant consistently outperforms all other methods across each dataset. Uncertainties indicate standard error over three runs.

|  | $\text{NFN}_{\text{HNP}}$ | $\text{NFN}_{\text{NP}}$ | DWSNets | MLP | $\text{MLP}_{\text{Aug}}$ |
|---|---|---|---|---|---|
| MNIST | $92.5 \pm 0.071$ | $\mathbf{92.9 \pm 0.218}$ | $74.4 \pm 0.143$ | $14.5 \pm 0.035$ | $21.0 \pm 0.172$ |
| FashionMNIST | $72.7 \pm 1.53$ | $\mathbf{75.6 \pm 1.07}$ | $64.8 \pm 0.685$ | $12.5 \pm 0.111$ | $15.9 \pm 0.181$ |
| CIFAR-10 | $44.1 \pm 0.471$ | $\mathbf{46.6 \pm 0.072}$ | $41.5 \pm 0.431$ | $16.9 \pm 0.250$ | $18.9 \pm 0.432$ |

Table 4: Classification test accuracies (%) for datasets of implicit neural representations (INRs) of either ShapeNet-10 [6] or ScanNet-10 [10]. Our equivariant NFNs outperform the MLP baselines and recent non-equivariant methods such as inr2vec [12]. Uncertainties indicate standard error over three runs.

|  | $\text{NFN}_{\text{HNP}}$ | $\text{NFN}_{\text{NP}}$ | MLP | $\text{MLP}_{\text{Aug}}$ | inr2vec[12] |
|---|---|---|---|---|---|
| ShapeNet-10 | $86.9 \pm 0.860$ | $\mathbf{88.7 \pm 0.461}$ | $25.4 \pm 0.121$ | $33.8 \pm 0.126$ | $39.1 \pm 0.385$ |
| ScanNet-10 | $64.1 \pm 0.572$ | $\mathbf{65.9 \pm 1.10}$ | $32.9 \pm 0.351$ | $45.5 \pm 0.126$ | $38.2 \pm 0.409$ |

original data directly from INR weights. Compared to discrete signal representations (pixels, voxels, etc.), the advantage of directly processing INRs is that one can easily be agnostic to varying signal sizes and resolutions.

In this task, our goal is to classify the contents of INRs given only the weights as input. We consider datasets of SIRENs [58] that encode images (MNIST [38], FashionMNIST [66], and CIFAR [35]) and 3D shapes (ShapeNet-10 and ScanNet-10 [53]). For image datasets each SIREN network represents the mapping from pixel coordinate to RGB (or grayscale) value for a single image, while for 3D shapes each network is a signed (or unsigned) distance function encoding a single shape. Each dataset of SIREN weights is split into training, validation, and testing sets. We construct and train invariant neural functionals to classify the INRs, and compare their performance against the MLP and $\text{MLP}_{\text{Aug}}$ baselines, which are three-layer MLPs with ReLU activations and 1,000 hidden units per layer.

As Navon et al. [47] test the performance of DWSNets on their independently constructed 2D-image INR datasets, we also present the results of training DWSNets on our own 2D-image INR datasets. Table 3 show that $\text{NFN}_{\text{NP}}$ consistently achieves the highest test accuracies of any method on the 2D-image tasks. More broadly, equivariant architectures significantly outperform the non-equivariant MLP approaches, even with permutation data augmentations.

For the 3D-shape datasets we also report the performance of inr2vec [12], a recent non-equivariant method for classifying 3D shapes from INR weights. Note that inr2vec's original setting assumes that all INRs in a dataset are trained from the same shared initialization, whereas our problem setting makes no such assumption and allows INRs to be trained from random and independent initializations. As expected, Table 4 shows that $\text{NFN}_{\text{HNP}}$ and $\text{NFN}_{\text{NP}}$ achieve significantly higher test accuracies than inr2vec, as well as the non-equivariant MLP baselines.

In addition to superior generalization, Appendix Table 16 shows that the NFNs are also better at fitting the training data compared to non-equivariant architectures. The MLPs achieve low train accuracy, even with an equal number of parameters as the NFNs. Interestingly, $\text{NFN}_{\text{NP}}$ matches or exceeds $\text{NFN}_{\text{HNP}}$ performance on both CIFAR-10 and the 3D-shape datasets while using fewer parameters (e.g., $35\%$ as many parameters on CIFAR-10). Finally, we note that no weight space methods (including NFN) match the performance of near state-of-the-art methods on discrete data representations such as 2D image arrays and point clouds (see Section E.2 for details). All weight space methods still lack the *geometric* inductive biases that, e.g., convolutional networks have in image tasks.

Table 5: Test accuracy (%) of training with winning tickets (95% sparsity masks) produced either by running IMP or predicted by an NFN. We also show the performance of Random ticket (random mask of equivalent sparsity level), and Dense training (no sparsity). We show results for MLPs (trained on MNIST) and CNNs (trained on CIFAR-10). Uncertainties show standard error over initializations.

| | Dense | IMP | Random | $\text{NFN}_{\text{NP}}$ | $\text{NFN}_{\text{PT}}$ |
|---|---|---|---|---|---|
| CIFAR-10 | $63.1 \pm 0.06$ | $44.0 \pm 0.06$ | $21.1 \pm 0.26$ | $\mathbf{41.4 \pm 0.08}$ | $\mathbf{42.6 \pm 0.07}$ |
| MNIST | $97.8 \pm 0.0$ | $96.2 \pm 0.04$ | $89.6 \pm 0.36$ | $\mathbf{94.8 \pm 0.01}$ | $\mathbf{95.0 \pm 0.01}$ |

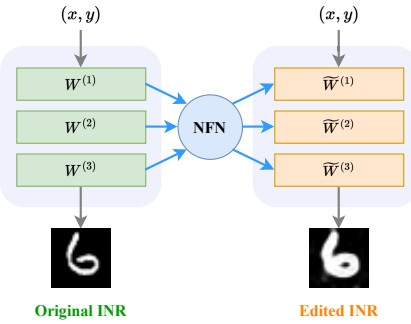

| Method | Contrast (CIFAR-10) | Dilate (MNIST) |
|---|---|---|
| MLP | 0.031 | 0.306 |
| $\text{MLP}_{\text{Aug}}$ | 0.029 | 0.307 |
| $\text{NFN}_{\text{PT}}$ | 0.029 | 0.197 |
| $\text{NFN}_{\text{HNP}}$ | **0.021** | **0.070** |
| $\text{NFN}_{\text{NP}}$ | **0.020** | **0.068** |

Figure 3: In weight-space style editing, an NFN directly edits the weights of an INR to alter the content it encodes. In this example, the NFN edits the weights to dilate the encoded image.

Table 6: Test mean squared error (lower is better) between weight-space editing methods and ground-truth image-space transformations.

### 3.3 Predicting "winning ticket" masks from initialization

The Lottery Ticket Hypothesis [19, 20, LTH] conjectures the existence of *winning tickets*, or sparse initializations that train to the same final performance as dense networks, and showed their existence in some settings through iterative magnitude pruning (IMP). IMP retroactively finds a winning ticket by pruning *trained* models by magnitude; however, finding the winning ticket from only the initialization without training remains challenging.

We demonstrate that permutation equivariant neural functionals are a promising approach for finding winning tickets at initialization by learning over datasets of initializations and their winning tickets. Let $U_0 \in \mathcal{U}$ be an initialization and let the sparsity mask $M \in \{0, 1\}^{\dim(\mathcal{U})}$ be a winning ticket for the initialization, with zeros indicating that the corresponding entries of $U_0$ should be pruned. The goal is to predict a winning ticket $\hat{M}$ given a held out initialization $U_0$, such that the MLP initialized with $U_0$ and sparsity pattern $\hat{M}$ will achieve a high test accuracy after training.

We construct a conditional variational autoencoder [32, 59, cVAE] that learns a generative model of the winning tickets conditioned on initialization and train on datasets of (initialization, ticket) pairs found by one step of IMP with a sparsity level of $P_m = 0.95$ for both MLPs trained on MNIST and CNNs trained on CIFAR-10. Table 5 compares the performance of tickets predicted by equivariant neural functionals against IMP tickets and random tickets. We generate random tickets by randomly sampling sparsity mask entries from Bernoulli$(1 - P_m)$. In this setting, $\text{NFN}_{\text{HNP}}$ is prohibitively parameter inefficient, but $\text{NFN}_{\text{NP}}$ is able to recover test accuracies that are close to that of IMP pruned networks in CIFAR-10 and MNIST, respectively. Somewhat surprisingly, $\text{NFN}_{\text{PT}}$ performs just as well as the other NFNs, indicating that one can approach IMP performance in these settings without considering interactions between weights or layers. Appendix E.4 further analyzes how $\text{NFN}_{\text{PT}}$ learns to prune.

### 3.4 Weight-space style editing

Another potentially useful application of neural functionals is to edit (i.e., transform) the weights of a given INR to alter the content that it encodes. In particular, the goal of this task is to edit the weights of a trained SIREN to alter its encoded image (Figure 3). We evaluate two editing tasks:

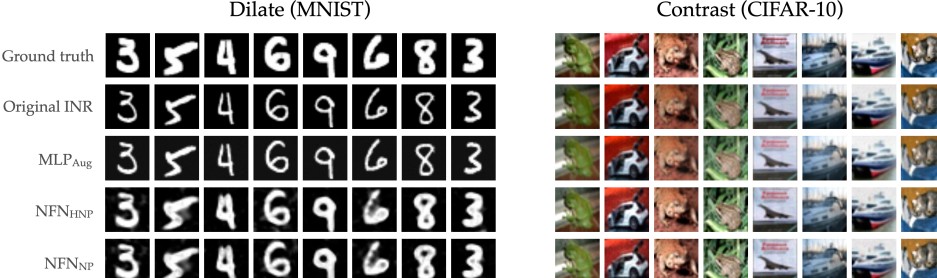

Figure 4: Random qualitative samples of INR editing behavior on the Dilate (MNIST) and Contrast (CIFAR-10) editing tasks. The first row shows the image produced by the original INR, while the rows below show the result of editing the INR weights with an NFN. The difference between MLP neural functionals and equivariant neural functionals is especially pronounced on the more challenging Dilate tasks, which require modifying the geometry of the image. In the Contrast tasks, the MLP baseline produces dimmer images compared to the ground truth, which is especially evident in the second and third columns.

(1) making MNIST digits thicker via image dilation (**Dilate**), and (2) increasing image contrast on CIFAR-10 (**Contrast**). Both of these tasks require neural functionals to process *the relationships between different pixels* to successfully solve the task.

To produce training data for this task, we use standard image processing libraries [28, OpenCV] to dilate or increase the contrast of the MNIST and CIFAR-10 images, respectively. The training objective is to minimize the mean squared error between the image generated by the NFN-edited INR and the image produced by image processing. We construct equivariant neural functionals to edit the INR weights, and compare them against MLP-based neural functionals with and without permutation augmentation.

Table 6 shows that permutation equivariant neural functionals (NFN$_{HNP}$ and NFN$_{NP}$) achieve significantly better test MSE when editing held out INRs compared to other methods, on both the Dilate (MNIST) and Contrast (CIFAR-10) tasks. In other words, they produce results that are closest to the "ground truth" image-space processing operations for each task. The pointwise ablation NFN$_{PT}$ performs significantly worse, indicating that accounting for interactions between weights and layers is important to accomplishing these tasks. Figure 4 shows random qualitative samples of editing by different methods below the original (pre-edit) INR. We observe that NFNs are more effective than MLP$_{Aug}$ at dilating MNIST digits and increasing the contrast in CIFAR-10 images.

# 4   Related work

The permutation symmetries of neurons have been a topic of interest in the context of loss landscapes and model merging [21, 4, 62, 17, 1]. Other works have analyzed the degree of learned permutation symmetry in networks that process weights [64] and studied ways of accounting for symmetries when measuring or encouraging diversity in the weight-space [13]. However, these symmetries have not been a key consideration in architecture design for processing weight-space objects [2, 39, 23, 36, 69, 13, 33]. Instead, existing approaches try to encourage permutation equivariance through data augmentation [51, 44]. In contrast, this work directly encodes the equivariance of the weight-space into our architecture design, which can result in much higher data and computational efficiency, as evidenced by the success of convolutional neural networks [37].

Our work follows a long line of literature that incorporates structure and symmetry into neural network architectures [37, 8, 54, 34, 9, 18]. This includes works that design permutation equivariant layers, originally for processing sets and graphs [52, 68, 25, 41]. More generally, equivariant layers have been developed for processing arbitrary rank-$k$ tensors (and correspondingly, hypergraphs) under higher-order permutation actions [40, 63, 48]. In this context, we can view feedforward networks as graphs that factor into a special layered structure where each (hidden) layer of neurons is an independently permutable set of nodes, and the weights are adjacency matrices specifying the edge weights between nodes of adjacent layers. Then our equivariant NF-Layers give the maximal set

of linear functions on these special graph structures. As discussed in Section 1, Navon et al. [47] recently developed an equivariant weight-space layer that is equivalent to our NF-Layer in the HNP setting. Our work introduces the NP setting to improve parameter efficiency and scalability over the HNP setting, and extends beyond the fully connected case to handle convolutional weight-space inputs.

## 5 Conclusion

This paper proposes a novel symmetry-inspired framework for the design of neural functional networks (NFNs), which process weight-space features such as weights, gradients, and sparsity masks. Our framework focuses on the permutation symmetries that arise in weight-spaces due to the particular structure of neural networks. We introduce two equivariant NF-Layers as building blocks for NFNs, which differ in their underlying symmetry assumptions and parameter efficiency, then use them to construct a variety of permutation equivariant neural functionals. Experimental results across diverse settings demonstrate that permutation equivariant neural functionals outperform prior methods and are effective for solving weight-space tasks.

**Limitations and future work.** Although we believe this framework is a step toward the principled design of effective neural functionals, there remain multiple directions for improvement. One such direction would concern extending the NF-Layers beyond feedforward weight spaces, in order to process the weights of more complex architectures such as ResNets [26] and Transformers [65]. Another useful direction would involve reducing the activation sizes produced by NF-Layers, in order to scale neural functionals to process the weights of very large networks. Finally, improvements to NF-Layers could account for the other symmetries of neural network weight spaces, such as scaling symmetries in ReLU networks [22].

## Acknowledgements

We thank Andy Jiang and Will Dorrell for helpful early discussions about theoretical concepts, and Yoonho Lee and Eric Mitchell for reviewing an early draft of this paper. AZ and KB are supported by the NSF Graduate Research Fellowship Program. This work was also supported by Apple.

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

# Appendix

# A  Equivariant NF-Layer pseudocode

Here we present a multi-channel implementation of the $\mathcal{S}$-equivariant NF-Layer presented in Eq. 2 (which ignores biases), using PyTorch [50] and Einops-like [55] pseudocode. That is, it implements a linear layer $H : \mathcal{W}^{c_i} \to \mathcal{W}^{c_o}$, where $c_i$ and $c_o$ are the number of input and output channels.

Note that our actual implementation differs from this pseudocode in a few ways: (1) it supports the full weight-space $\mathcal{U}$ which includes biases, (2) it supports convolution weights as well as fully connected weights, and (3) it initializes parameters based on the fan-in of the NF-Layer, instead of from $\mathcal{N}(0, 1)$.

```python
class NPLayer(nn.Module):
  def __init__(self, L, co, ci):
      super().__init__()
      # initialize weights. co=output channels, ci=input channels.
      self.A = nn.Parameter(torch.randn(L, L, co, ci))
      self.B = nn.Parameter(torch.randn(L, co, ci))
      self.B_prev = nn.Parameter(torch.randn(L, co, ci))
      self.C = nn.Parameter(torch.randn(L, co, ci))
      self.C_next = nn.Parameter(torch.randn(L, co, ci))
```

```
10          self.D = nn.Parameter(torch.randn(L, co, ci))
11
12      def forward(self, W):
13          # Input W is a list of L weight-space tensors with shapes:
14          #      [(B, c_i, n_1, n_0), ..., ((B, c_i, n_L, n_{L-1}))]
15          # We return a list of L tensors with shapes:
16          #      [(B, c_o, n_1, n_0), ..., ((B, c_o, n_L, n_{L-1}))]
17          # where c_i, c_o are the input and output channels.
18
19          # Compute [W_{\star,:}^{(1)}, \cdots, W_{\star,:}^{(L)}]. Each has shape (B, c_i, n_{i-1}).
20          row_means = [w.mean(-2) for w in W]
21          # Compute [W_{:,\star}^{(1)}, \cdots, W_{:,\star}^{(L)}]. Each has shape (B, c_i, n_i).
22          col_means = [w.mean(-1) for w in W]
23          # Compute [W_{\star,\star}^{(1)}, \cdots, W_{\star,\star}^{(L)}] with shape (B, c_i, L).
24          rowcol_means = torch.stack([w.mean(dim=(-2, -1)) for w in W], -1)
25          out_W = []
26          for i, (w, a, b, b_prev, c, c_next, d) in enumerate(zip(
27              W, self.A, self.B, self.B_prev, self.C, self.C_next, self.D
28          )):
29              # Calculates \sum_s a^{i,s} W_{\star,\star}^{(s)}.
30              h1 = einsum(a, rowcol_means, 'co ci s, B ci s -> B co () ()')
31              # Calculates b^{i,i} W_{\star,k}^{(i)}.
32              h2 = einsum(b, row_means[i], 'co ci, b ci n_im1 -> B co () n_im1')
33              if i > 0:
34                  # Calculates b^{i,i-1} W_{k,\star}^{(i-1)}.
35                  h2 += einsum(b_prev, col_means[i-1], 'co ci, B ci n_im1 -> B co () n_im1')
36              # Calculates c^{i,i} W_{j,\star}^{(i)}.
37              h3 = einsum(c, col_means[i], 'co ci, B ci n_i -> b co n_i ()')
38              if i < self.L - 1:
39                  # Calculates c^{i,i+1} W_{\star,j}^{(i+1)}.
40                  h3 += einsum(c_next, row_col[i+1], 'co ci, B ci n_i -> B co n_i ()')
41              # Calculates d^i W_{jk}^{(i)}.
42              h4 = einsum(d, w, 'co ci, B ci n_i n_im1 -> B co n_i n_im1')
43              # Calculates H(W^{(i)}) with shape (B, c_o, n_i, n_{i-1}).
44              out_W.append(h1 + h2 + h3 + h4)
45          return out_W
```

# B    NF-Layers for the NP Setting

This section presents futher details related to the $\mathcal{S}$-equivariant layer $H : \mathcal{U} \to \mathcal{U}$ and the $\mathcal{S}$-invariant layer $P$. First, Sec. B.1 gives the full definition of $H$ as a function of both weight-space and bias-space features. Sec. B.2 provides a proof that $H(\cdot)$ expresses any $\mathcal{S}$-equivariant linear function on $\mathcal{U}$ (Proposition 1). Sec. B.3 proves that combining our invariant layer $P$ with a fully connected layer can express any $\mathcal{S}$-invariant function from $\mathcal{U}$ to $\mathbb{R}^d$ (Proposition 2).

## B.1    Full definition

We present the full definition of the $\mathcal{S}$-equivariant NF-Layer $H : \mathcal{U} \to \mathcal{U}$, which completes Eq. 2 by also including the biases. The layer processes a set of weights $U = (W, v)$ and outputs arrays $(Y, z)$

Table 7: $\mathcal{S}$-equivariant parameter sharing for linear maps $\mathrm{vec}(U) \mapsto \theta\mathrm{vec}(U)$. Parameter sharing is a system of constraints on the entries of $(\vartheta, \phi, \varphi, \psi) = \theta$. Each table is organized by the layer indices $(i, s)$. For example, the first table says that for any $(i, s)$ where $s = i - 1$, we constrain $\vartheta_{spq}^{ijk} = \vartheta_{s,p',q'}^{i,j',k'} = a_\vartheta^{i,i-1}$ for any $j, k, p, q$ and $j', k', p', q'$ where $k \neq p$ and $k' \neq p'$. After parameter sharing, we observe that there are only a constant number of free parameters for each $(i, s)$ pair, adding up to $O\left(L^2\right)$ parameters total.

|  | $s = i - 1$ | | $s = i$ | | $s = i + 1$ | | other $s$ |
|---|---|---|---|---|---|---|---|
| $\vartheta_{spq}^{ijk}$ | $\begin{cases} a_\vartheta^{i,i-1} & k \neq p \\ b_\vartheta^{i,i-1} & k = p \end{cases}$ | | $\begin{cases} a_\vartheta^{i,i} & j \neq p, k \neq q \\ b_\vartheta^{i,i} & j = p, k \neq q \\ c_\vartheta^{i,i} & j \neq p, k = q \\ d_\vartheta^{i} & j = p, k = q \end{cases}$ | | $\begin{cases} a_\vartheta^{i,i+1} & j \neq q \\ c_\vartheta^{i,i+1} & j = q \end{cases}$ | | $a_\vartheta^{i,s}$ |

|  | $s = i - 1$ | | $s = i$ | | other $s$ |
|---|---|---|---|---|---|
| $\phi_{sp}^{ijk}$ | $\begin{cases} a_\phi^{i,i-1} & k \neq p \\ b_\phi^{i,i-1} & k = p \end{cases}$ | | $\begin{cases} a_\phi^{i,i} & j \neq p \\ b_\phi^{i,i} & j = p \end{cases}$ | | $a_\phi^{i,s}$ |

|  | $s = i$ | | $s = i + 1$ | | other $s$ |
|---|---|---|---|---|---|
| $\varphi_{spq}^{ij}$ | $\begin{cases} a_\varphi^{i,i-1} & j \neq p \\ b_\varphi^{i,i-1} & j = p \end{cases}$ | | $\begin{cases} a_\varphi^{i,i} & j \neq q \\ b_\varphi^{i,i} & j = q \end{cases}$ | | $a_\varphi^{i,s}$ |

|  | $s = i$ | | other $s$ |
|---|---|---|---|
| $\psi_{sp}^{ij}$ | $\begin{cases} a_\psi^{i,i} & j \neq p \\ b_\psi^{i} & j = p \end{cases}$ | | $a_\psi^{i,s}$ |

with the same dimensions. The learnable parameters are in blue:

$$H(U) := (Y, z) \in \mathcal{W} \times \mathcal{V} \tag{3}$$

$$Y_{jk}^{(i)} := \sum_s \left( a_\vartheta^{i,s} W_{\star,\star}^{(s)} + a_\phi^{i,s} v_\star^{(s)} \right) + b_\vartheta^{i,i} W_{\star,k}^{(i)} + b_\vartheta^{i,i-1} W_{k,\star}^{(i-1)}$$

$$+ b_\phi^i v_j^{(i)} + c_\vartheta^{i,i} W_{j,\star}^{(i)} + c_\vartheta^{i,i+1} W_{\star,j}^{(i+1)} + c_\phi^i v_k^{(i-1)} + d_\vartheta^i W_{jk}^{(i)}$$

$$z_j^{(i)} := \sum_s \left( a_\varphi^{i,s} W_{\star,\star}^{(s)} + a_\psi^{i,s} v_\star^{(s)} \right) + b_\varphi^{i,i} W_{j,\star}^{(i)} + b_\varphi^{i,i+1} W_{\star,j}^{(i+1)} + b_\psi^i v_j^{(i)},$$

where $\star$ denotes summation or averaging over a dimension.

## B.2 Proof of Proposition 1

Here we show that the layer defined in Eq. 3 is not only $\mathcal{S}$-equivariant, it can express *every* linear equivariant function $\mathcal{U} \to \mathcal{U}$. Our strategy is to first consider general linear layers $T(\cdot; \theta) : \mathcal{U} \to \mathcal{U}$ parameterized by $\theta \in \Theta$, and then constrain to $\theta$ such that $T(\cdot; \theta)$ is $\mathcal{S}$-equivariant. Applying this constraint will directly yield Eq. 3.

For our purposes, it is sometimes convenient to distinguish layer, row, and column indices of entries in $U$, so we split the parameters $\theta = (\vartheta, \phi, \varphi, \psi)$ and write $T : \mathcal{U} \to \mathcal{U}$ in the form:

$$T(U; \theta) := (Y(U), z(U)) \quad \in \mathcal{W} \times \mathcal{V} = \mathcal{U} \tag{4}$$

$$Y(U)_{jk}^{(i)} := \sum_{s=1}^L \sum_{p=1}^{n_s} \sum_{q=1}^{n_{s-1}} \vartheta_{spq}^{ijk} W_{pq}^{(s)} + \sum_{s=1}^L \sum_{p=1}^{n_s} \phi_{sp}^{ijk} v_p^{(s)} \tag{5}$$

$$z(U)_j^{(i)} := \sum_{s=1}^L \sum_{p=1}^{n_s} \sum_{q=1}^{n_{s-1}} \varphi_{spq}^{ij} W_{pq}^{(s)} + \sum_{s=1}^L \sum_{p=1}^{n_s} \psi_{sp}^{ij} v_p^{(s)}. \tag{6}$$

We introduce the notation $U_\alpha$ to identify individual entries of $U$. Here $\alpha$ is a tuple of length two or three, for indexing into either a weight or bias. We denote the space of valid index tuples of $\mathcal{W}$ and $\mathcal{V}$ by $\mathbb{W}$ and $\mathbb{V}$, respectively, and define $\mathbb{U} := \mathbb{W} \cup \mathbb{V}$ as the combined index space of $\mathcal{U}$. For example, if $\alpha = (i, j, k) \in \mathbb{W}$, then $U_\alpha = W_{jk}^{(i)}$.

We can then define the index space $\mathbb{I} := \mathbb{U} \times \mathbb{U}$ for parameters $\theta \in \Theta$. We use $\theta_\beta^\alpha$ to index an entry of $\theta$ with upper and lower indices $[\alpha, \beta] \in \mathbb{I}$. For example, if $\alpha = (i, j, k) \in \mathbb{W}$ and $\beta = (s, p, q) \in \mathbb{W}$, we have $\theta_\beta^\alpha = \vartheta_{spq}^{ijk}$.

By flattening $U$ into a vector, we can think of $\theta$ as a matrix with indices $\alpha$ and $\beta$. Eq. 4 can be rewritten in the flattened form:

$$T(U; \theta)_\alpha = \sum_\beta \theta_\beta^\alpha U_\beta. \tag{7}$$

In the flattened representation, we can write the action of $\sigma \in \mathcal{S}$ on $\text{vec}(U)$ by a permutation matrix $P_\sigma$. Equivariance requires that $P_\sigma \theta \text{vec}(U) = \theta P_\sigma \text{vec}(U)$ for any $\sigma \in \mathcal{S}$. Since the input $U$ can be anything, we get the following constraint on $\theta$:

$$P_\sigma \theta = \theta P_\sigma, \quad \forall \sigma \in \mathcal{S}. \tag{8}$$

We can re-express the action of $\mathcal{S}$ on $\mathcal{U}$ (Eq. 1) as an action on the index space $\mathbb{U}$. For $\alpha \in \mathbb{U}$, we write $\sigma(\alpha)$ which is defined:

$$\sigma(i, j, k) = (i, \sigma_i(j), \sigma_{i-1}(k)) \quad (i, j, k) \in \mathbb{W} \tag{9}$$
$$\sigma(i, j) = (i, \sigma_i(j)) \quad (i, j) \in \mathbb{V}, \tag{10}$$

for any $\sigma \in \mathcal{S}$. We extend this definition into an action of $\mathcal{S}$ on $\mathbb{I}$:

$$\sigma[\alpha, \beta] := [\sigma\alpha, \sigma\beta], \quad [\alpha, \beta] \in \mathbb{U} \times \mathbb{U}. \tag{11}$$

We can now rewrite the equivariance constraint in Eq. 8 using indices $\alpha, \beta$:

$$[P_\sigma \theta]_\beta^\alpha = \theta_\beta^{\sigma^{-1}(\alpha)} = \theta_{\sigma(\beta)}^\alpha = [\theta P_\sigma]_\beta^\alpha. \tag{12}$$

By relabeling $\alpha \leftarrow \sigma^{-1}(\alpha)$, we can rewrite this condition $\theta_\beta^\alpha = \theta_{\sigma(\beta)}^{\sigma(\alpha)}$. Hence, equivariance is equivalent to requiring that $\theta$ share parameters within orbits under the action of $\mathcal{S}$ on its indices $\alpha, \beta$ [54, Prop 3.1]. We characterize these orbits in Sec. B.2.1, yielding Table 7 which gives the parameter sharing as a system of constraints on the indices of $\theta$. These constraints characterize the space of $\mathcal{S}$-equivariant $T(\cdot, \theta)$. In Sec. B.2.2, we conclude the proof by showing that under parameter sharing, $T(\cdot, \theta)$ is equivalent to Eq. 3.

### B.2.1 Parameter orbits and sharing

We now partition the parameters of $\theta$ into orbits under the $\mathcal{S}$-action on its index space, in order to generate the parameter sharing constraints given in Table 7. The index space of $\theta$ is $\mathbb{I} = \mathbb{U} \times \mathbb{U}$. There are four subsets of $\mathbb{I}$:

1. $\mathbb{I}^{WW} := \mathbb{W} \times \mathbb{W}$: Contains $[\alpha, \beta] = [(i, j, k), (s, p, q)]$, indexing parameters $\vartheta_{spq}^{ijk}$.

2. $\mathbb{I}^{WV} := \mathbb{W} \times \mathbb{V}$: Contains $[\alpha, \beta] = [(i, j, k), (s, p)]$, indexing parameters $\phi_{sp}^{ijk}$.

3. $\mathbb{I}^{VW} := \mathbb{V} \times \mathbb{W}$: Contains $[\alpha, \beta] = [(i, j), (s, p, q)]$, indexing parameters $\varphi_{spq}^{ij}$.

4. $\mathbb{I}^{VV} := \mathbb{V} \times \mathbb{V}$: Contains $[\alpha, \beta] = [(i, j), (s, p)]$, indexing parameters $\psi_{sp}^{ij}$.

Consider the block of indices $\mathbb{I}^{WW} = \mathbb{W} \times \mathbb{W}$, containing $[\alpha, \beta] = [(i, j, k), (s, p, q)]$ indexing parameters $\vartheta_\beta^\alpha$. Since the $\mathcal{S}$-action never changes the layer indices $(i, s)$, we can independently consider orbits within sub-blocks of indices $\mathbb{I}_{i,s}^{WW} = \{ [(i, j, k), (s, p, q)] \mid \forall j, k, p, q \}$. The number of orbits within each sub-block $\mathbb{I}_{i,s}^{WW}$ depends on the relationship between the layer indices $i$ and $s$: they are either the same layer ($s = i$), they are adjacent ($s = i - 1$ or $s = i + 1$), or they are non-adjacent ($s \notin \{i - 1, i, i + 1\}$). We now analyze the orbits of sub-blocks for a few cases.

If $s = i - 1$, then choose any two indices $\left[\alpha^{(1)}, \beta^{(1)}\right], \left[\alpha^{(2)}, \beta^{(2)}\right] \in \mathbb{I}_{i,s}^{WW}$ where the first satisfies $p \neq k$ and the second satisfies $p = k$. Then the orbits of each index are:

$$\text{Orbit}\left(\left[\alpha^{(1)}, \beta^{(1)}\right]\right) = \{\, [(i, j, k), (s, p, q)] \mid \forall j, k, p, q : p \neq k \,\} \tag{13}$$

$$\text{Orbit}\left(\left[\alpha^{(2)}, \beta^{(2)}\right]\right) = \{\, [(i, j, k), (s, p, q)] \mid \forall j, k, p, q : p = k \,\}. \tag{14}$$

We see that these two orbits actually partition the entire sub-block of indices $\mathbb{I}_{i,s}^{WW}$, with each orbit characterized by whether or not $p = k$. We introduce the parameters $a_\vartheta^{i,i-1}$ (for the first orbit) and $b_\vartheta^{i,i-1}$ (for the second orbit). Under equivariant parameter sharing, all parameters of $\vartheta$ corresponding $\mathbb{I}_{i,s}^{WW}$ are equal to either $a_\vartheta^{i,i-1}$ or $b_\vartheta^{i,i-1}$, depending on whether $p = k$ or $p \neq k$.

If $s = i + 1$, we instead choose any two indices where the first satisfies $j \neq q$ and the second satisfies $j = q$. Then the sub-block of indices $\mathbb{I}_{i,s}^{WW}$ is again partitioned into two orbits:

$$\{\, [(i, j, k), (s, p, q)] \mid \forall j, k, p, q : j \neq q \,\}, \text{ and } \{\, [(i, j, k), (s, p, q)] \mid \forall j, k, p, q : j = q \,\} \tag{15}$$

depending on the condition $j = q$. We name two parameters $a_\vartheta^{i,i+1}$ and $c_\vartheta^{i,i+1}$ for this sub-block, with one for each orbit.

We can repeat this process for sub-blocks of $\mathbb{I}^{WW}$ where $i = s$ and $s \notin \{i - 1, i, i + 1\}$, as well as for the other three blocks of $\mathbb{I}$. Table 7 shows the complete parameter sharing constraints on $\theta$ resulting from partitioning all possible sub-blocks into orbits.

**Number of parameters.** We also note that every layer pair $(i, s)$ introduces only a constant number of parameters: the number of parameters in each cell of Table 7 has no dependence on the input, output, or hidden dimensions of $\mathcal{U}$. Hence the number of distinct parameters after parameter sharing simply grows with the number of layer pairs, i.e. $O\left(L^2\right)$.

### B.2.2 Equivalence to equivariant NF-Layer definition

All that remains is to show that the map $T(\cdot; \theta) : \mathcal{U} \to \mathcal{U}$ with $\mathcal{S}$-equivariant parameter sharing (Table 7) is equivalent to the NF-Layer $H$ we defined in Eq. 3.

Consider a single term from Eq. 4 where $s = i - 1$. Substituting using the constraints of Table 7, we simplify:

$$\sum_{p,q} \vartheta_{i-1,p,q}^{i,j,k} W_{pq}^{(i-1)} = a^{i,i-1} \sum_q \sum_{k \neq p} W_{p,q}^{(i-1)} + b^{i,i-1} \sum_q W_{k,q}^{(i-1)} \tag{16}$$

$$= a^{i,i-1} W_{\star,\star}^{(i-1)} + (b^{i,i-1} - a^{i,i-1}) W_{k,\star}^{(i-1)}.$$

We can then reparameterize $b^{i,i-1} \leftarrow b^{i,i-1} - a^{i,i-1}$, resulting in two terms that appear in Eq. 3. We can simplify every term of Eq. 4 in a similar manner using the parameter sharing of Table 7, reducing the general layer to the $\mathcal{S}$-equivariant NF-Layer.

### B.3 Proof of Proposition 2

Analogous to Sec. B.2, we begin with general parameterized linear maps $T(\cdot; \theta) : \mathcal{U} \to \mathbb{R}^d$ and characterize the parameter sharing implied by $\mathcal{S}$-invariance. We then show that $T(\cdot, \theta)$ under these constraints is equivalent to $f \circ P$, where $f : \mathbb{R}^{2L} \to \mathbb{R}^d$ is a parameterized linear map (i.e., a fully connected layer) and $P(U) = \left(W_{\star,\star}^{(1)}, \cdots, W_{\star,\star}^{(L)}, v_\star^{(1)}, \cdots, v_\star^{(L)}\right)$. We will assume $d = 1$ for the rest of this section, but it is straightforward to generalize our arguments to arbitrary $d$.

In general, we can parameterize linear maps $\mathcal{U} \to \mathbb{R}$ by $\theta \in \mathbb{R}^{\dim(\mathcal{U})}$, where:

$$T(U, \theta) = \sum \theta_\alpha U_\alpha, \tag{17}$$

where $\alpha$ is an index for either weights or biases as in Sec. B.2. The invariance constraint $T(U) = T(\sigma U)$ implies:

$$\theta_\alpha = \theta_{\sigma(\alpha)}, \tag{18}$$

where $\sigma(\alpha)$ is defined by Eq. 9.

We can now characterize the orbits of the indices of $\theta$ under $\mathcal{S}$. In this case, the orbits (and resulting parameter sharing) are relatively simple. For a fixed $i$, all $(i, j, k) \in \mathbb{W}$ form an orbit, and we introduce a shared parameter $a^i$. Similarly, for each $i$ all $(i, j) \in \mathbb{V}$ form an orbit, and we introduce a shared parameter $b^i$ for each. We can then rewrite the layer:

$$T(U; \theta) = \sum_i \left( a^i \sum_{jk} W^{(i)}_{jk} + b^i \sum_j v^{(i)}_j \right) = \sum_i a^i W^{(i)}_{\star,\star} + b^i v^{(i)}_\star. \tag{19}$$

This is simply a linear combination of the output of $P(U)$ parameterized by $\{(a^i, b^i)\}$, and can be written as the composition $f \circ P$ where $f$ is a fully connected layer.

## C NF-Layers for the HNP setting

### C.1 Equivariant NF-Layer

Because an expression for the $\tilde{\mathcal{S}}$-equivariant NF-Layer analogous to Eq. 3 would be unwieldy, we instead define the layer in terms of its parameter sharing (Tables 8-11) on $\theta$.

We can derive HNP-equivariant parameter sharing of $\theta$ using a similar strategy to Sec. B.2.1: we partition the index spaces $\mathbb{I}^{WW}, \mathbb{I}^{WV}, \mathbb{I}^{VW}, \mathbb{I}^{VV}$ into orbits under the action of $\tilde{\mathcal{S}}$, and share parameters within each corresponding orbit of $\vartheta, \phi, \varphi, \psi$. The resulting parameter sharing is different from the NP-setting because while the action of $\mathcal{S}$ on $\mathcal{U}$ could permute the rows and columns of every weight and bias, the action of $\tilde{\mathcal{S}}$ on $\mathcal{U}$ does not affect the columns of $W^{(1)}$ or the rows of $W^{(L)}, v^{(L)}$, which correspond to input and output dimensions (respectively).

The orbits are again analyzed within sub-blocks defined by the values of the layer indices $(i, s)$. As with the NP setting, there are broadly four types of sub-blocks based on whether $i = s$, $s = i - 1$, $s = i + 1$, or $s \notin \{i - 1, i, i + 1\}$. However, there are now additional considerations based on whether $i$ or $s$ is an input or output layer. For example, consider the sub-block of $\mathbb{I}^{WW}$ where $i = s = 1$, which we denote $\mathbb{I}^{WW}_{1,1}$. The action on the indices in this sub-block can be written $\sigma[\alpha, \beta] = [(1, \sigma_1(j), k), (1, \sigma_1(p), q)]$. Importantly, the column indices $k, q$ are never permuted since they correspond to the input layer. We see that $\mathbb{I}^{WW}_{1,1}$ contains two orbits *for each* $k \in [\![1..n_0]\!]$ and $q \in [\![1..n_0]\!]$, with the two orbits characterized by whether or not $j = p$. Hence we have $2n_0^2$ orbits and Table 8 introduces $2n_0^2$ parameters $\left\{ a^{1,1,k,q}_\vartheta, b^{1,1,k,q}_\vartheta \mid k, q \in [\![1..n_0]\!] \right\}$ for this sub-block of parameters.

Now consider another sub-block of $\mathbb{I}^{WW}$ where $1 < i = s < L$. Now the action of $\tilde{\mathcal{S}}$ on indices in this sub-block can be written $\sigma[\alpha, \beta] = [(i, \sigma_i(j), \sigma_{i-1}(k)), (i, \sigma_i(p), \sigma_{i-1}(q))]$. Then we have a total of two orbits characterized by whether or not $k = p$, rather than $2n_0^2$ orbits for the $i = 1$ case. Tables 8-11 present the complete parameter sharing for each of $\vartheta, \phi, \varphi, \psi$, resulting from analyzing every possible orbit within any sub-block of $\mathbb{I}^{WW}, \mathbb{I}^{WV}, \mathbb{I}^{VW}, \mathbb{I}^{VV}$.

### C.2 Invariant NF-Layer

While the NP-invariant NF-Layer sums over the rows and columns of every weight and bias, under HNP assumptions there is no need to sum over the columns of $W^{(1)}$ (inputs) or the rows of $W^{(L)}, v^{(L)}$ (outputs). So the HNP invariant NF-Layer $\tilde{P} : \mathcal{U} \to \mathbb{R}^{2L+n_0+2n_L}$ is defined:

$$\tilde{P}(U) = \left( P(U), W^{(1)}_{\star,:}, W^{(L)}_{:,\star}, v^{(L)} \right), \tag{20}$$

where $W^{(1)}_{\star,:}$ and $W^{(L)}_{:,\star}$ denote summing over only the rows or only the columns of the matrix, respectively. Note that $\tilde{P}$ satisfies $\tilde{\mathcal{S}}$-invariance without satifying $\mathcal{S}$-invariance.

## D Additional experimental details

### D.1 Predicting generalization

The model we use consists of three equivariant NF-Layers with 16, 16, and 5 channels respectively. We apply ReLU activations after each linear NF-Layer. The resulting weight-space features are passed

Table 8: HNP-equivariant parameter sharing on $\vartheta \subseteq \theta$, corresponding to the NF-Layer $\tilde{H} : \mathcal{U} \to \mathcal{U}$.

$$\vartheta^{ijk}_{spq}$$

| | $i=2$ | $2<i<L$ | $i=L$ |
|---|---|---|---|
| $s=i-1$ | $\begin{cases} a^{2,1,q}_\vartheta & k\neq p \\ b^{2,1,q}_\vartheta & k=p \end{cases}$ | $\begin{cases} a^{i,i-1}_\vartheta & k\neq p \\ b^{i,i-1}_\vartheta & k=p \end{cases}$ | $\begin{cases} a^{L,L-1,j}_\vartheta & k\neq p \\ b^{L,L-1,j}_\vartheta & k=p \end{cases}$ |

| | $i=1$ | $1<i<L$ | $i=L$ |
|---|---|---|---|
| $s=i$ | $\begin{cases} a^{1,1,k,q}_\vartheta & j\neq p \\ b^{1,1,k,q}_\vartheta & j=p \end{cases}$ | $\begin{cases} a^{i,i}_\vartheta & j\neq p, k\neq q \\ b^{i,i}_\vartheta & j=p, k\neq q \\ c^{i,i}_\vartheta & j\neq p, k=q \\ d^{i,i}_\vartheta & j=p, k=q \end{cases}$ | $\begin{cases} a^{L,L,j,p}_\vartheta & k\neq q \\ c^{L,L,j,p}_\vartheta & k=q \end{cases}$ |

| | $i=1$ | $1<i<L-1$ | $i=L-1$ |
|---|---|---|---|
| $s=i+1$ | $\begin{cases} a^{1,2,k}_\vartheta & j\neq q \\ c^{1,2,k}_\vartheta & j=q \end{cases}$ | $\begin{cases} a^{1,2}_\vartheta & j\neq q \\ c^{1,2}_\vartheta & j=q \end{cases}$ | $\begin{cases} a^{L-1,L,p}_\vartheta & j\neq q \\ c^{L-1,L,p}_\vartheta & j=q \end{cases}$ |

| | $i=1, 1<s<L$ | $i=1, s=L$ | $1<i<L, s=L$ |
|---|---|---|---|
| other $s$ | $a^{1,s,k}_\vartheta$ | $a^{1,L,k,p}_\vartheta$ | $a^{i,L,p}_\vartheta$ |
| | $1<i<L, s=1$ | $i=L, s=1$ | $i=L, 1<s<L$ |
| | $a^{i,1,q}_\vartheta$ | $a^{L,1,j,q}_\vartheta$ | $a^{L,s,j}_\vartheta$ |
| | $1<i<L, 1<s<L$ | | |
| | $a^{i,s}_\vartheta$ | | |

Table 9: HNP-equivariant parameter sharing on $\phi \subset \theta$, corresponding to the NF-Layer $\tilde{H} : \mathcal{U} \to \mathcal{U}$.

$$\phi^{ijk}_{sp}$$

| | | $1<i<L$ | $i=L$ |
|---|---|---|---|
| $s=i-1$ | | $\begin{cases} a^{i,i-1}_\phi & k\neq p \\ b^{i,i-1}_\phi & k=p \end{cases}$ | $\begin{cases} a^{L,L-1,j}_\phi & k\neq p \\ b^{L,L-1,j}_\phi & k=p \end{cases}$ |

| | $i=1$ | $1<i<L$ | $i=L$ |
|---|---|---|---|
| $s=i$ | $\begin{cases} a^{1,1,k}_\phi & j\neq p \\ b^{1,1,k}_\phi & j=p \end{cases}$ | $\begin{cases} a^{i,i}_\phi & j\neq p \\ b^{i,i}_\phi & j=p \end{cases}$ | $b^{L,L,j,p}_\phi$ |

| | $i=1, 1<s<L$ | $i=1, s=L$ | $1<i<L, s=L$ |
|---|---|---|---|
| other $s$ | $a^{1,s,k}_\phi$ | $a^{1,L,k,p}_\phi$ | $a^{i,L,p}_\phi$ |
| | $1<i<L, 1\leq s<L$ | $i=L, 1\leq s<L$ | |
| | $a^{i,s}_\phi$ | $a^{L,s,j}_\phi$ | |

Table 10: HNP-equivariant parameter sharing on $\varphi \subset \theta$, corresponding to the NF-Layer $\tilde{H} : \mathcal{U} \to \mathcal{U}$.

| $\varphi^{ij}_{spq}$ | $i=1$ | $1<i<L$ | $i=L$ |
|---|---|---|---|
| $s=i$ | $\begin{cases}a_\varphi^{1,1,q} & j\neq p\\ b_\varphi^{1,1,k} & j=p\end{cases}$ | $\begin{cases}a_\varphi^{i,i} & j\neq p\\ b_\varphi^{i,i} & j=p\end{cases}$ | $b_\varphi^{L,L,j,p}$ |
| $s=i+1$ | | $1\leq i<L-1$ | $i=L-1$ |
| | | $\begin{cases}a_\varphi^{i,i+1} & j\neq p\\ b_\varphi^{i,i+1} & j=p\end{cases}$ | $\begin{cases}a_\varphi^{L-1,L,p} & j\neq p\\ b_\varphi^{L-1,L,p} & j=p\end{cases}$ |
| other $s$ | $1\leq i<L, s=1$ | $1\leq i<L, 1<s<L$ | $1\leq i<L, s=L$ |
| | $a_\varphi^{i,s,q}$ | $a_\varphi^{i,s}$ | $a_\varphi^{i,L,p}$ |
| | $i=L, s=1$ | $i=L, 1<s<L$ | |
| | $a_\varphi^{L,1,j,q}$ | $a_\varphi^{L,s,j}$ | |

Table 11: HNP-equivariant parameter sharing on $\psi \subset \theta$, corresponding to the NF-Layer $\tilde{H} : \mathcal{U} \to \mathcal{U}$.

| $\psi^{ij}_{sp}$ | | $1\leq i<L$ | $i=L$ |
|---|---|---|---|
| $s=i$ | | $\begin{cases}a_\psi^{i,i} & j\neq p\\ b_\psi^{i,i} & j=p\end{cases}$ | $b_\psi^{L,L,j,p}$ |
| other $s$ | $1\leq i<L, s=L$ | $i=L, 1\leq s<L$ | $1\leq i<L, 1\leq s<L$ |
| | $a_\psi^{i,L,p}$ | $a_\psi^{L,s,j}$ | $a_\psi^{i,s}$ |

into an invariant NF-Layer with mean pooling. The output of the invariant NF-Layer is flattened and projected to $\mathbb{R}^{1,000}$. The resulting vector is then passed through an MLP with two hidden layers, each with 1,000 units and ReLU activations. The output is linearly projected to a scalar and passed through a sigmoid function. Since the output of the model can be interpreted as a probability, we train the model with binary cross-entropy with hyperparameters outlined in Table 12. The model is trained for 50 epochs with early stopping based on $\tau$ on the validation set, which takes 1 hour on a Titan RTX GPU.

Table 12: Hyperparameters for predicting generalization on Small CNN Zoo.

| Name | Values |
|---|---|
| Optimizer | Adam |
| Learning rate | 0.001 |
| Batch size | 8 |
| Loss | Binary cross-entropy |
| Epoch | 50 |

## D.2 Predicting "winning ticket" masks from initialization

Concretely, the encoder learns the posterior distribution $q_\theta(Z \mid U_0, M)$ where $Z \in \mathbb{R}^{\dim(\mathcal{U}) \times C}$ is the latent variable for the winning tickets and $C$ is the number of latent channels. The decoder learns $p_\theta(M \mid U_0, Z)$, and both encoder and decoder are implemented using our equivariant NF-Layers. For the prior $p(Z)$ we choose the isometric Gaussian distribution, and train using the evidence lower bound (ELBO):

$$\mathcal{L}_\theta(M, U_0) = \mathbb{E}_{z \sim q_\theta(\cdot \mid U_0, M)}\Big[ \ln p_\theta(M \mid U_0, z) \Big] - \mathrm{D}_{\mathrm{KL}}\Big( q_\theta(\cdot \mid U_0, M) \,\|\, p(\cdot) \Big).$$

The initialization and sparsity mask are concatenated so the input to the encoder $q_\theta$ is $(U, M) \in \mathbb{R}^{\dim(\mathcal{U}) \times 2}$. After the bottleneck, we concatenate the latent variables and the original mask along the channels, i.e. the decoder input is $(U_0, Z) \in \mathbb{R}^{\dim(\mathcal{U}) \times (C+1)}$.

The first dataset uses three-layer MLPs with 128 hidden units trained on MNIST and the second uses CNNs with three convolution layers (128 channels) and 2 fully-connected layers trained on CIFAR-10. In each dataset, we include 400 pairs for training and hold out 50 for evaluation. The hyperparameter details are in Table 13. The encoder and decoder models contain 4 equivariant NF-Layers with 64 hidden channels within each layer. The latent variable is 5 dimensions. Training takes 5H on a Titan RTX GPU.

Table 13: Hyperparameters for predicting LTH on MNIST and CIFAR-10.

| Name | Values |
|---|---|
| Optimizer | Adam |
| Learning rate | $1 \times 10^{-3}$ |
| Batch size | [4, 8] |
| Epoch | 200 |

### D.3 Classifying INRs

We use SIREN [58] for our INRs of CIFAR-10, FashionMNIST, and MNIST. For the SIREN models, we used a three-layer architecture with 32 hidden neurons in each layer. We trained the SIRENs for 5,000 steps using Adam optimizer with a learning rate of $5 \times 10^{-5}$. Datasets were split into 45,000 training images, 5,000 validation images, and 10,000 (MNIST, CIFAR-10) or 20,000 (FashionMNIST) test images. We trained 10 copies (MNIST, FashionMNIST) or 20 copies (CIFAR-10) of SIRENs on each training image with different initializations, and a single SIREN on each validation and test image. No additional data augmentation was applied. For 3D shape classification, we adopt the same protocol introduced in [12], and we train each SIREN to fit the Unsigned Distance Function (UDF) value of points sampled around a shape. Each SIREN is composed of a single hidden layer with 128 neurons. We use Adam as an optimizer and we train for 1,000 steps.

We also trained neural functionals with three equivariant NF-Layers + ReLU activations, each with 512 channels, followed by invariant NF-Layers (mean pooling) and a three-layer MLP head with 1,000 hidden units and ReLU activation. Dropout was applied to the MLP head only. For the NFN IO-encoding, we used sinusoidal position encoding with a maximum frequency of 10 and 6 frequency bands (dimension 13). The training hyperparameters are shown in Table 14, and training took $\sim$ 4H on a Titan RTX GPU.

Table 14: Hyperparameters for classifying INRs on MNIST and CIFAR-10 using neural functionals.

| Name | Values |
|---|---|
| Optimizer | Adam |
| Learning rate | $1 \times 10^{-4}$ |
| Batch size | 32 |
| Training steps | $2 \times 10^5$ |
| MLP dropout | 0.5 |

We also experimented with larger MLPs (4,000 and 8,000 hidden units per layer) that have parameter counts comparable to those of the NFNs, but found that it did not significantly increase test accuracy, as shown in Table 16.

### D.4 Weight-space style editing

For weight-space editing, we use the same INRs as the ones used for classification but we do not augment the dataset with additional INRs. Let $U_i$ be the INR weights for the $i^{\text{th}}$ image and $\text{SIREN}(x, y; U)$ be the output of the INR parameterized by $U$ at coordinates $(x, y)$. We edit the INR

weights $U'_i = U_i + \gamma \cdot \text{NFN}(U_i)$, and $\gamma$ is a learned scalar initialized to 0.01. Letting $f_i(x, y)$ be the pixel values of the ground truth edited image (obtained from image-space processing), the objective is to minimize mean squared error:

$$\mathcal{L}(\text{NFN}) = \frac{1}{N \cdot d^2} \sum_{i=1}^{N} \sum_{x,y}^{d} \|\text{SIREN}(x, y; U') - f_i(x, y)\|_2^2. \tag{21}$$

Note that since the SIREN itself is differentiable, the loss can be directly backpropagated through $U'$ to the parameters of the NFN.

The neural functionals contain 3 equivariant NF-Layers with 128 channels, one invariant NF-Layer (mean pooling) followed by 4 linear layers with 1,000 hidden neurons. Every layer uses ReLU activation. The training hyperparameters can be found in Table 15, and training takes $\sim 1$ hour on a Titan RTX GPU.

Table 15: Hyperparameters for weight-space style editing using neural functionals.

| Name | Values |
|---|---|
| Optimizer | Adam |
| Learning rate | $1 \times 10^{-3}$ |
| Batch size | 32 |
| Training steps | $5 \times 10^4$ |

# E    Additional experiments and analysis

## E.1    Analyzing train vs test accuracies in INR classification

Table 16 shows the training and test accuracies of our NFNs and other weight space methods across the benchmarks we consider in this paper. An interesting observation is that in addition to better test performance, NFN typically achieve much higher *training* accuracies compared to the MLPs. $\text{MLP}_{\text{Aug}}$, the MLP variant trained with permutation augmentations, especially struggles to achieve low training error. This indicates that training architectures to respect neuron permutation symmetries is extremely challenging, perhaps because these symmetry groups are extremely large [1]. Therefore, encoding equivariance or invariance via architecture design, as we do with NFNs, is much more effective.

## E.2    Non-weight-space baselines for image classification

For the image and 3D shape classification tasks discussed in Sec. 3.2, we compared the neural functionals to non-weight-space baselines, namely residual networks [26] for MNIST, FashionMNIST, and CIFAR-10, and PointNet [52] for ShapeNet-10 and ScanNet-10. The results, shown in Table 16, indicate that applying state of the art architectures to pixels or point clouds is more effective than any weight space method. This may be because these architectures can leverage geometric properties of image and shape classification tasks. While neural functionals handle weight space symmetries, it remains an open problem to incorporate the geometric symmetries of common machine learning problems into weight space methods.

## E.3    Ablating positional encoding

We ablate the effect of IO-encoding on the $\text{NFN}_{\text{NP}}$ architectures, in the 2D-INR classification and style editing tasks. The results, reported in Tables 17 and 18, show that IO-encoding adds a very small (sometimes negligible) boost to $\text{NFN}_{\text{NP}}$ performance, though it never hurts. Since $\text{NFN}_{\text{NP}}$ often performs as well as or better than $\text{NFN}_{\text{HNP}}$, this indicates that even the base NP variant can solve many weight-space tasks without needing to break that symmetry.

## E.4    Interpreting learned lottery ticket masks

We further analyze the behavior of $\text{NFN}_{\text{PT}}$ on lottery ticket mask prediction by plotting the mask score predicted for a given initialization value at each layer. To make the visualization clear we train

Table 16: Classification train and test accuracies (%) on datasets of 2D INRS (MNIST, Fashion-MNIST, and CIFAR-10) and 3D INRs (ShapeNet-10 and ScanNet-10). Our equivariant NFNs outperform the MLP baselines, even when the MLPs have permutation augmentations to encourage invariance, and even when the MLPs are significantly larger to match NFN's parameter counts (MLP-4000/8000). NFNs also outperform recent non-equivariant methods developed for 3D INRs such as inr2vec [12]. ResNet-18 and PointNet operate on gridded or pointcloud signals, instead of weight space representations, and their performance is given for reference. PointNet results were obtained directly from Qi et al. [52]. Where given, uncertainties indicate standard error over three runs.

| | | MNIST | FashionMNIST | CIFAR-10 | ShapeNet-10 | ScanNet-10 |
|---|---|---|---|---|---|---|
| $\text{NFN}_{\text{HNP}}$ | Train | $94.9 \pm 0.579$ | $82.3 \pm 2.78$ | $75.5 \pm 0.810$ | $100 \pm 0.0$ | $100.0 \pm 0.0$ |
| | Test | $92.5 \pm 0.071$ | $72.7 \pm 1.53$ | $44.1 \pm 0.471$ | $86.9 \pm 0.860$ | $64.1 \pm 0.572$ |
| $\text{NFN}_{\text{NP}}$ | Train | $95.0 \pm 0.115$ | $81.8 \pm 0.868$ | $66.0 \pm 0.694$ | $100.0 \pm 0.0$ | $100.0 \pm 0.0$ |
| | Test | $\mathbf{92.9 \pm 0.218}$ | $\mathbf{75.6 \pm 1.07}$ | $\mathbf{46.6 \pm 0.072}$ | $\mathbf{88.7 \pm 0.461}$ | $\mathbf{65.9 \pm 1.10}$ |
| MLP | Train | $42.4 \pm 2.44$ | $44.5 \pm 2.17$ | $23.7 \pm 2.39$ | $100.0 \pm 0.0$ | $100.0 \pm 0.0$ |
| | Test | $14.5 \pm 0.035$ | $12.5 \pm 0.111$ | $16.9 \pm 0.250$ | $25.4 \pm 0.121$ | $32.9 \pm 0.351$ |
| $\text{MLP}_{\text{Aug}}$ | Train | $20.5 \pm 0.401$ | $14.9 \pm 1.45$ | $19.1 \pm 1.75$ | $34.0 \pm 0.0$ | $42.7 \pm 0.012$ |
| | Test | $21.0 \pm 0.172$ | $15.9 \pm 0.181$ | $18.9 \pm 0.432$ | $33.8 \pm 0.126$ | $45.5 \pm 0.126$ |
| MLP-4000 | Train | $72.6 \pm 1.39$ | – | $30.4 \pm 0.521$ | – | – |
| | Test | $15.5 \pm 0.090$ | – | $17.1 \pm 0.120$ | – | – |
| $\text{MLP-4000}_{\text{Aug}}$ | Train | $19.4 \pm 1.39$ | – | $20.5 \pm 0.333$ | – | – |
| | Test | $21.1 \pm 0.010$ | – | $19.3 \pm 0.325$ | – | – |
| MLP-8000 | Train | $77.8 \pm 1.74$ | – | $35.9 \pm 0.868$ | – | – |
| | Test | $15.8 \pm 0.014$ | – | $17.3 \pm 0.280$ | – | – |
| $\text{MLP-8000}_{\text{Aug}}$ | Train | $20.3 \pm 3.30$ | – | $18.1 \pm 0.347$ | – | – |
| | Test | $21.3 \pm 0.075$ | – | $19.6 \pm 0.060$ | – | – |
| inr2vec[12] | Train | – | – | – | $99.0 \pm 0.0$ | $93.8 \pm 0.090$ |
| | Test | – | – | – | $39.1 \pm 0.385$ | $38.2 \pm 0.409$ |
| ResNet-18 | Test | $99.21 \pm 0.28$ | $95.48 \pm 0.13$ | $93.95 \pm 0.08$ | – | – |
| PointNet | Test | – | – | – | $94.3$ | $72.7$ |

Table 17: Classification test accuracies (%) when ablating the positional encoding for implicit neural representations (INRs) of MNIST, FashionMNIST, and CIFAR-10. Uncertainties indicate standard error over three runs.

| | $\text{NFN}_{\text{NP}}$ | $\text{NFN}_{\text{NP}}$ (no positional encoding) |
|---|---|---|
| MNIST | $92.9 \pm 0.218$ | $92.9 \pm 0.077$ |
| CIFAR-10 | $46.6 \pm 0.072$ | $46.5 \pm 0.160$ |
| FashionMNIST | $75.6 \pm 1.07$ | $73.4 \pm 0.701$ |

Table 18: Test mean squared error when ablating the positional encoding on the Dilate (MNIST) and Contrast (CIFAR-10) implicit neural representation (INR) editing task. Uncertainties indicate standard error over three runs.

| | $\text{NFN}_{\text{NP}}$ | $\text{NFN}_{\text{NP}}$ (no positional encoding) |
|---|---|---|
| Contrast (CIFAR-10) | $0.020 \pm 0.000$ | $0.020 \pm 0.000$ |
| Dilate (MNIST) | $0.068 \pm 0.000$ | $0.070 \pm 0.001$ |

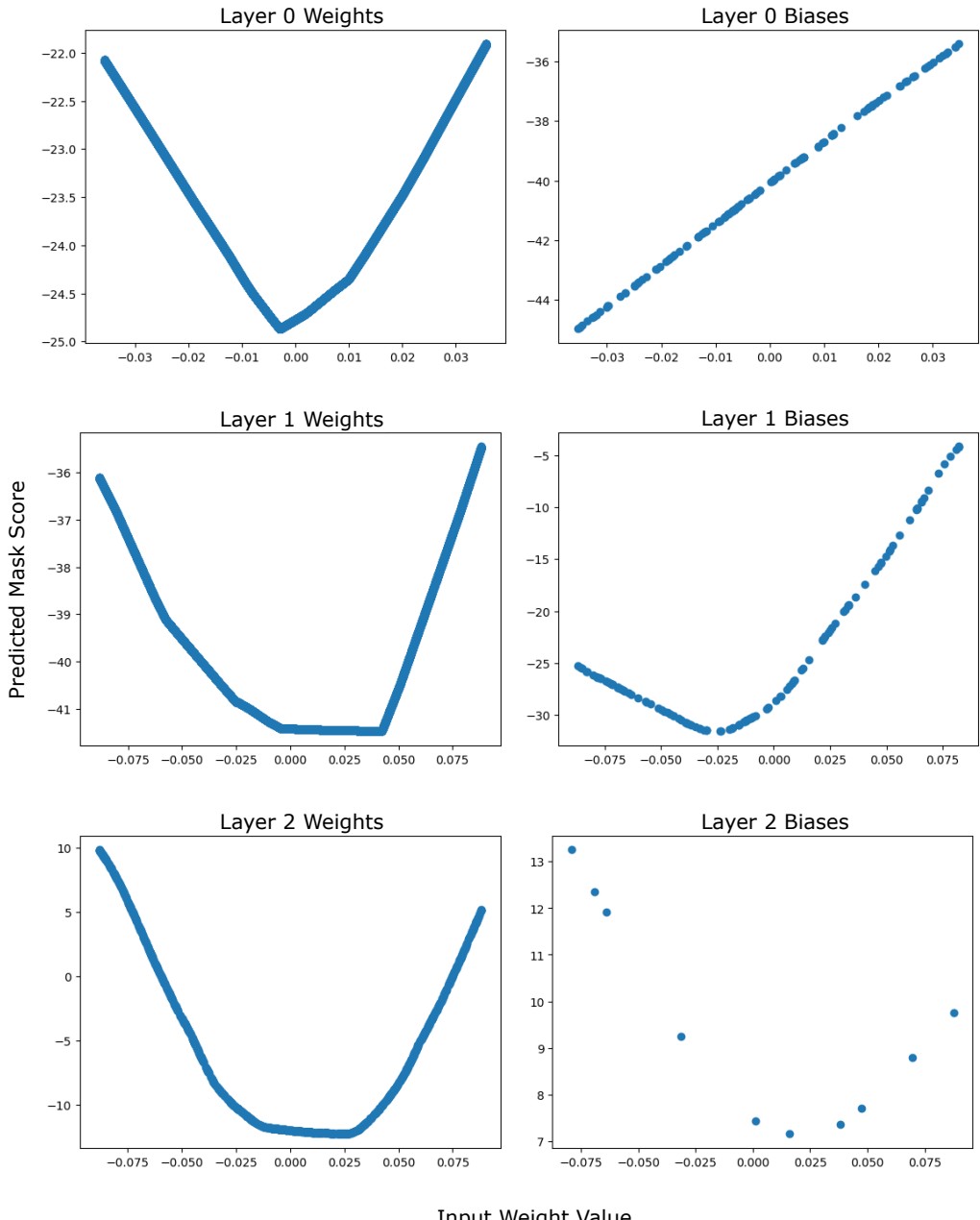

Figure 5: Mask scores vs weight magnitude for modified NFN$_{\text{PT}}$.

NFN$_{\text{PT}}$ on MLP mask prediction without layer norm, which can be viewed as a scalar function of the initialization $f^{(i)} : \mathbb{R} \to \mathbb{R}$ for each layer $i$. Figure 5 plots, for a fixed latent value, the predicted mask score as a function of the initialization value (low mask scores are pruned, while high mask scores are not). These plots suggest that, in the MLP setting, neural functionals are learning something similar to magnitude pruning of the initialization. In our setting, this turns out to be a strong baseline for lottery ticket mask prediction: the test accuracy of models pruned with the modified network is 95.0%.

Table 19: Kendall's $\tau$ coefficient and $R^2$ between predicted and actual test accuracies of three- and five-layer MLPs trained on MNIST. Our equivariant neural functionals outperform the baseline from [64] which predicts generalization using only simple weight statistics as features. Uncertainties indicate standard error over five runs.

|  |  | NFN$_{\text{HNP}}$ | NFN$_{\text{NP}}$ | STATNN |
|---|---|---|---|---|
| $\tau$ | 3-Layer | **0.876 ± 0.003** | 0.859 ± 0.002 | 0.854 ± 0.002 |
|  | 5-Layer | **0.871 ± 0.001** | 0.855 ± 0.001 | 0.860 ± 0.001 |
| $R^2$ | 3-Layer | **0.957 ± 0.003** | 0.9424 ± 0.003 | 0.937 ± 0.002 |
|  | 5-Layer | **0.956 ± 0.002** | 0.947 ± 0.001 | 0.950 ± 0.001 |

### E.5 Predicting MLP generalization from weights

In addition to predicting generalization on the Small CNN Zoo benchmark (Section 3.1), we also construct our own datasets to evaluate predicting generalization on MLPs. Specifically, we study three- and five-layer MLPs with 128 units in each hidden layer. For each of the two architectures, we train 2,000 MLPs on MNIST with varying optimization hyperparameters, and save 10 randomly-selected checkpoints from each run to construct a dataset of 20,000 (weight, test accuracy) pairs. Runs are partitioned according to a 90% / 10% split for training and testing.

We evaluate NFN$_{\text{HNP}}$ and NFN$_{\text{NP}}$ on this task and compare them to the STATNN baseline [64] which predicts test accuracy from hand-crafted features extracted from the weights. Table 19 shows that NFN$_{\text{NP}}$ and STATNN are broadly comparable, while NFN$_{\text{HNP}}$ consistently outperform other methods across both datasets in two measures of correlation: Kendall's tau and $R^2$. These results confirm that processing the raw weights with permutation equivariant neural functionals can lead to greater predictive power when assessing generalization from weights.

