# OpenReview forum: "Permutation Equivariant Neural Functionals"
_NeurIPS.cc/2023/Conference — NeurIPS 2023 poster_

### Official Review · Reviewer_7rfP · 2023-06-22

**Soundness:** 3 good
**Presentation:** 4 excellent
**Contribution:** 3 good
**Rating:** 7
**Confidence:** 4

**Summary:**

The paper introduces and evaluates permutation equivariant neural functional networks (NFNs).
Neural functionals are models which take weights of other neural networks, or more general weight-space features, like gradients or sparsity masks, as inputs.
Their permutation equivariance addresses the issue that any particular vector of weight-space features is just an arbitrary representative of an equivalence class of weight-space features that correspond to exactly the same network.
Specifically, the weights and biases in all hidden layers may by permuted arbitrarily, which corresponds to simultaneous permutations of rows and columns of the previous and following weight matrix, respectively.
Permutation equivariant NFNs ensure predictions which are independent from the particular permutation.
As usual for equivariant networks, this reduces the number of model parameters (of the NFN) significantly, and makes learning feasible in the first place.

The authors consider two specific group actions, which are the first order permutation actions and trivial actions on invariant scalars.
Equivariance refers throughout the paper to mappings that commute with the first order action in the input and output, while invariance refers to mappings from first order actions to scalars.
Equation 3 presents a linear NFN layer, which computes features as usual by taking learned linear combinations of weight-space features that are summed over different combinations of rows and columns.
Proposition 1 claims that this layer is 1) sufficient and 2) necessary for equivariance, i.e. spans the space of the most general linear equivariant maps (I have some concerns here, discussed in the weakness section).
As nonlinearities, NFNs use pointwise nonlinearities, which commute with permutation actions.
A straightforward extension to permutations of CNN channels is discussed towards the end of section 2.2.

Besides the HNP model, which considers the "correct" hidden neuron permutation symmetry,
an NP model, which additionally permutes input and output neurons and has a further reduced number of parameters, is introduced.
In order for the network to be able to break this excess equivariance, positional encodings are added.

Extensive experiments evaluate HPN-NFNs and NP-NFNs and compare them to baseline models.
The first two applications are permutation invariant predictions of CNN test accuracies from their weights and invariant classifications of implicit neural representations.
In both cases the NFNs perform significantly better than baselines, and the HNP variant wins over the NP task in the first but loses in the second one (they are still quite close).
The other two applications are permutation equivariant predictions of a lottery ticket hypothesis winning ticket sparsity mask and the editing of implicit neural representations.
In the former, NP-NFN performs close to a task-specific baseline, while HNP-NFN is prohibitively expensive, thus supporting the need for the NP-NFN model.
In the latter, both NP-NFN and HNP-NFN outperform their baselines.

**Strengths:**

The paper is well motivated and very clearly written - it was really a joy to read!
After introducing the mathematical framework, the authors derive the most general linear equivariant mappings for the considered group actions (I have some concerns here, see the weakness section).
Explicit equations for the required number of parameters of the different models are given.
The experiments are very extensive and show the utility and superiority of NFNs in various applications.

**Weaknesses:**

I have two technical concerns regarding the claims about the generality of the proposed linear equivariant map in proposition 1.
Firstly, it should be mentioned that the proof assumes a specific choice of group action w.r.t. which the layer is equivariant, namely first order permutation actions.
Permutation equivariant networks have also been built for other actions, like irreducible representations or higher order tensor product actions - these would lead to other layers.
This point needs to be discussed and the statement of proposition 1 needs to be made precise, e.g. by saying that the map $T$ is equivariant w.r.t. the specific first order action $\sigma$ in the input and output.

[edit: this claim was wrong]
~Secondly, I believe that the mapping in equation 3 is only then the most general $S_{n_1}\times\dots\times S_{n_{L-1}}$ equivariant intertwiner when $n_1\neq\dots\neq n_{L-1}$ is assumed.
If, however, $n_i=n_j$ for some layers $i\neq j$, the same permutation group $S_{n_i} = S_{n_j}$ acts on these neurons, and it should be possible to linearly accumulate the corresponding weight-space features after $\star$-summing over the other indices.
A similar summing over features is already present in equation 3, namely the first term, which sums invariant scalars that are accumulated from all layers.
Note that this contribution can in practice become very large, since networks are in practice often constructed such that they have the same number of hidden neurons or channels throughout many different layers.
The more general equivariant mappings should be benchmarked against the current NFNs in the experimental section.~

The sufficiency and necessity of the invariant NFN layers in section 2.3. is not supported by a similar theorem as proposition 1.

Due to these weaknesses I decided to only give a weak accept to the current state.
However, the paper has a huge potential and I would be happy to switch to a strong accept when the issues are addressed.

**Questions:**

The paper is very clearly written and I don't have any remaining questions.


**Limitations:**

Some limitations and potential future work are discussed in the conclusion.
It should be added that one could consider more general permutation group actions.

---

> ### Author Rebuttal · Authors · 2023-08-10
>
> Thank you for your review and detailed analysis of the technical aspects of our work.
>
> > It should be mentioned that the proof assumes a specific choice of group action w.r.t. which the layer is equivariant, namely first order permutation actions.
>
> We will make it clear at the beginning of the proof that we are only concerned with equivariance to a particular choice of action (as defined in Equation 1).
>
> > Secondly, I believe that the mapping in equation 3 is only then the most general $S\_{n\_1} \times \cdots \times S\_{n\_{L-1}}$ equivariant intertwiner when $n\_1 \neq \cdots \neq n\_{L-1}$ is assumed.
>
> We don't follow how two layers having the same number of neurons creates a special case. To aid us in understanding and responding to this point better, could you write out an example of any additional or missing terms explicitly, which would show that Eq 3 is not fully general?
>
> Additionally, would this claim not contradict Proposition 1? Though it is possible that we made a mistake and dropped terms somewhere in the derivation of the layer.
>
> > The sufficiency and necessity of the invariant NFN layers in section 2.3. is not supported by a similar theorem as proposition 1.
>
> We will add an analogous proposition as Prop 1, but for the sufficiency and necessity of the invariant layers in Section 2.3, using a similar strategy as in Appendix B.2.

---

> > ### Comment · Reviewer_7rfP · 2023-08-11
> >
> > Dear Authors, thank you for your replies.
> >
> > Instead of mentioning the group action in the proof, could you please mention it in the theorem? This should really be part of the statement itself. I would also strongly encourage a paragraph on other group actions, e.g. mentioning that they exist, might have a different performance, and citing related work.
> >
> > My reasoning in the second weakness was indeed erroneous: I was thinking about an action of the _diagonal subgroup_ $S_N$ of $S_N \times S_N$ for two layers with $N$ neurons, which would allow to linearly combine their features. This is of course not the case, since the factors act independently.
> >
> > Thanks for adding the proposition.
> >
> > I didn't read the concurrent/prior work by Navon et al. My review was and is only concerning the content of the current submission, irrespective of whether there was a similar submission a few months earlier.
> >
> > I am generally happy with the submission. If the authors address the first point in this reply I would update my rating to 7.

---

> > > ### Author Response · Authors · 2023-08-11
> > >
> > > Thank you for your prompt response.
> > >
> > > > Instead of mentioning the group action in the proof, could you please mention it in the theorem? This should really be part of the statement itself.
> > >
> > > Understood. We will modify both Proposition (1) and the preliminaries to make this clear. We will modify L91-93 (introducing S-equivariance) as follows:
> > >
> > > "We refer to $f$ as **S-equivariant** if $\sigma f(U) = f(\sigma U)$ for all $\sigma \in \mathcal{S}, U \in \mathcal{U}$. In this work, we exclusively focus on equivariance with respect to the action introduced above so, e.g., $\sigma U$ is defined according to Eq (1)."
> > >
> > > We will then modify Proposition (1) as follows:
> > >
> > > "The NF-Layer $H$ (Eq 3-4) is S-equivariant with respect to the action in Eq (1), applied to both the input and output spaces. [...]"
> > >
> > > > I would also strongly encourage a paragraph on other group actions, e.g. mentioning that they exist, might have a different performance, and citing related work.
> > >
> > > We will include this--for higher order actions we assume you are referring to the line of work in, e.g., [1,2]? We will also discuss the line of work approaching equivariant layer design by focusing on irreducible representations. We also welcome any references you think may be relevant and will include and discuss them.
> > >
> > > [1] Thiede et al. The general theory of permutation equivarant neural networks and higher order graph variational encoders.
> > >
> > > [2] Pan and Kondor. Permutation Equivariant Layers for Higher Order Interactions.
> > >
> > > >  I was thinking about an action of the _diagonal subgroup_ […]
> > >
> > > Understood, thanks for clarifying. We will also modify the preliminaries to make it clear that the $\sigma_i$ are independent.

---

> > > > ### Comment · Reviewer_7rfP · 2023-08-12
> > > >
> > > > Great, I updated my rating to 7.
> > > >
> > > > Yes, these are the papers I referred to.

---

### Official Review · Reviewer_9eY1 · 2023-06-23

**Soundness:** 3 good
**Presentation:** 3 good
**Contribution:** 3 good
**Rating:** 7
**Confidence:** 2

**Summary:**

This paper proposes the NF-Layer that maps the weight space of a deep neural network (DNN), including MLP and CNN, to another weight space, possibly with a different number of channels. Neural Functional Networks (NFNs) are then constructed using the NL-Layers to process the weight space of a DNN. NF-Layer is designed to be S-equivariant, that is, if the input weights of a DNN are permutated so as not to change the output of the DNN, the output of the NF-Layer, which is also in another weight space, is also permutated in the same order. Such a property enjoys the equivariant characteristics inherent in the DNN weights and enhances the efficiency in modeling NFNs, like the convolution structure enhances the efficiency of modeling image inputs. Experimental results are presented to demonstrate the usefulness of NF-Layers in applications of NFNs, including accuracy prediction, classification of implicit neural representations (INRs), winning ticket prediction, and style-editing through INRs.


**Strengths:**

* An interesting application of the equivariant architecture based on parameter-sharing.
* Introduction of the NP setting in the permutation definition. Although it requires a stronger assumption than the HNP setting, the resulting NF-Layer can be modeled more efficiently under the NP assumption.
* Experiments suggesting the advantage of the proposed NF-Layers in interesting applications of NFNs.


**Weaknesses:**

* A network architecture with equivariance based on parameter-sharing is not original and has been proposed in [51], as the authors suggest.
* NF-Layers cannot directly handle some DNN models, including ResNet and Transformer.

(Minor)
* Figure 1 is difficult to understand at first glance.


**Questions:**

I can understand the permutation equivariant property in DNN weights, but I cannot intuitively understand why the prediction accuracy of NFNs improves with NF-Layers with S-equivariance. Is it because the parameter to be trained is reduced by the S-equivariant structure, while retaining its expressiveness?

It would be better to explain the multi-channel extension of permutations more carefully in ll.86-90. When I first read it, I mistakenly thought it was referring to the convolution channel. Also, it is better to describe that the proposed NF-Layer may change the channel size in this sense in the explanation of Figure 1.

In Figure 2, $W_{\ast,j}^{(i+1)}$ (with yellow c) -> $W_{j,\ast}^{(i)}$?

Equation in Figure 2 should be matched with Eq.(3).

In ll.155-160, I could not understand the motivation to use position embedding with NFN_{NP}. In which experiments with NFN_{NP}, position embeddings are used?

**Limitations:**

As described by the authors, there is a concurrent work by Navon et al.[45] that has a very similar motivation and methodology. But I think this paper has its original contributions that the authors mention and deserves to be published.

---

> ### Author Rebuttal · Authors · 2023-08-10
>
> Thank you for your review and interesting question.
>
> > A network architecture with equivariance based on parameter-sharing is not original and has been proposed in [51]
>
> To clarify, [51] (Equivariance through parameter sharing) provides general strategies for developing layers equivariant to a given choice of group action. [51] does not study the permutation symmetries of neural networks' weights and does not develop ways of processing weights. Rather, we utilize strategies from [51] to develop our NF-layers.
>
> > NF-Layers cannot directly handle some DNN models, including ResNet and Transformer.
>
> This is a good observation, architectures like ResNet have more complicated weight space permutation symmetries compared to feedforward networks so our definitions of the (hidden) neuron permutation groups do not directly apply. We believe that extending these layers to handle more general network topologies is an important (and nontrivial) direction of future work.
>
> > cannot intuitively understand why the prediction accuracy of NFNs improves with NF-Layers with S-equivariance. Is it because the parameter to be trained is reduced by the S-equivariant structure, while retaining its expressiveness?
>
> You are correct that enforcing S-equivariance reduces the parameter space, and we know that the target function must be S-equivariant (or invariant). So ideally enforcing equivariance does not sacrifice our ability to express the target function, while reducing the space of parameters to search over, leading to better generalization.
>
> > It would be better to explain the multi-channel extension of permutations more carefully in ll.86-90. When I first read it, I mistakenly thought it was referring to the convolution channel.
>
> Thanks for your suggestions–we will update our presentation of the NF-layers to more clearly explain that they can have arbitrary input and output channels, and avoid any confusion with channels in, e.g., the convolutional weight space being processed.
>
> > Possible error in figure 2?
>
> Thanks for pointing out this error, we will update the figure to fix this.
>
> > In ll.155-160, I could not understand the motivation to use position embedding with NFN_{NP}. In which experiments with NFN_{NP}, position embeddings are used?
>
> We use positional embeddings (PE) in all of our experiments. The reason is that our tasks (and most real world tasks involving weights) do not allow arbitrary permutations of the input and output layers of the networks being processed--only the hidden layers can be permuted. So the NP assumptions give an incorrect symmetry, which can be broken by PE.
>
> By analogy, Transformers are equivariant to token permutations, but most sequence-to-sequence tasks are not actually permutation symmetric (order matters). So Transformers often use PE to break that symmetry.
>
> We will update L155-160 to clarify this point.
>
> ## References
> [1] Cohen and Welling. Group Equivariant Convolutional Networks.

---

> > ### Comment · Reviewer_9eY1 · 2023-08-11
> > **Thank you for your replies**
> >
> > Thank you for your replies. I understand the relationship between [51] and this paper. I would like to keep my score.

---

> > > ### Author Response · Authors · 2023-08-11
> > >
> > > Acknowledged, thank you for reading our response and for your prompt reply!

---

### Official Review · Reviewer_jZvx · 2023-06-29

**Soundness:** 4 excellent
**Presentation:** 4 excellent
**Contribution:** 3 good
**Rating:** 7
**Confidence:** 5

**Summary:**

This paper studies the problem of defining linear layers (and by extension, neural networks) that operate on neural network weight spaces. The core idea of this work is to take into account weight permutation symmetries, similar to Navon et al., ICML’23. In particular, the weights of certain feedforward architectures, such as MLPs and CNNs, can be permuted in several ways, without altering the function that the neural net represents. Therefore, the authors seek to define functions on the weights, that will be equivariant (or invariant) to these very symmetries.

To achieve this, they characterise the space of linear equivariant/invariant functions, following the framework of Ravanbakhsh et al., ICML’17, i.e. by identifying the parameters of these functions that should be shared. The authors propose two variants, one that is derived from the detailed characterisation of the weight permutation symmetries that arise from hidden neurons ($\text{NFN}_{\text{HNP}}$), and another one that assumes extra permutation symmetries in the input/output layer (that can be broken with positional encodings), and is, therefore, more parameter efficient.
Interestingly, these results easily carry over from MLPs to CNNs. The proposed layers are experimentally tested in a battery of tasks, such as predicting NN generalisation or sparsity masks for NN pruning, showing considerable improvement against baselines that do not take into account permutation symmetries, in addition to a heavily reduced parameter count.


**Strengths:**

**Significance**: The proposed method has wide applicability, as correctly pointed out by the authors in the first paragraph of the introduction, and can potentially have a substantial impact on multiple neural-network-related problems (meta-learning, neural network editing, INR processing etc.).

**Presentation, clarity and reproducibility**: Although the concepts studied in this paper and the neural network architectures resulting from the study of the relevant symmetries, have a fair amount of complexity and may require a good understanding of the study of symmetries, the authors have made a very good effort to make their paper as accessible as possible. In particular, the illustrations, the nice summarisation of the core layer in Eq. (3) as well as the parameter counts in Table (1) and the accompanying explanations provide a good overview of the method that I believe will be appreciated by future readers. Moreover, the authors have formulated their layers in such a way that makes them easy to implement, which is also apparent from the pseudocode provided in the appendix.

**Experimental evidence**. The authors have tested their layers against a diverse set of problems and the reported results clearly motivate the need for equivariant/invariant layers and support all the design choices made.

**Extended scope**: A few months ago, another work by Navon et al., ICML’23 characterised the same family of equivariant/invariant layers as well (providing additional theoretical support). This is obviously not a problem, since the two works were developed in parallel. It is therefore nice to see that the present paper has a bonus, i.e. an extended scope compared to the aforementioned one since (1) the formulation seamlessly allows to extend the method to CNNs, and (2) the NP formulation allows for more practical implementation, a significant reduction in the number of parameters, and empirically strong performance. (3) This is probably subjective and a matter of taste, but I appreciated the alternative way to derive the layers (see sec B.3. in the appendix), using the parameter-sharing framework of Ravanbakhsh et al., ICML’17. I found it easier to follow and slightly more intuitive, which might be of independent technical interest in the future. Therefore, although the layers have been rediscovered, I feel that the contributions are still valuable.

**Weaknesses:**

**Clarity of the proofs (appendix)**: I found the explanations in section B.4. a bit hard to follow. Personally, I am familiar with the work of Navon et al., so I could easily understand the concepts, but I fear a reader not versed in the topic, might get lost in this part. Since it is crucial in order to gain a deep understanding of the paper, I would recommend that the authors try to simplify some concepts (e.g. by giving more details for a particular sub-case).

**Related work**: I believe that the authors do not give enough credit to the work by Navon et al.. As I mentioned before, the two works are concurrent, and they apparently independently discovered the same core idea, but I think that the authors should be more upfront about this and mention this early on in the paper and cite Navon et al. more prominently.


**Questions:**

**Experiments, baselines and implementation details**: Most of the following questions ask for clarifications and I do not consider them as weaknesses. However, I think a discussion is needed and it would be useful to include the answers to the below in the paper. In detail:

- **NP case**.  What are the positional encodings that are used in the experimental section? If I understand correctly they are fixed (handcrafted). Have the authors tried learnable ones instead? What is the number of added parameters when using learnable positional encodings? How do the authors explain the fact that using positional encodings in NFN_NP performs better than NFN_HNP?
- **NFN_PT case**. Since this method performs well in the pruning mask prediction experiment, why didn’t the authors also include its performance in the rest of the experiments (e.g. generalisation and INR classification)?
- **Pruning**: Why did the authors use a generative model (CVAE) for the winning ticket experiment? Why not a deterministic predictor (a 0-1 classifier)? Have the authors tested this? How far are the generated sparsity masks from the ones that IMP yields? Why isn’t the performance of MLP and MLP_aug included in this table?
- In section 3.2, would it be possible for completeness to add some baselines that classify images and 3D shapes, using different representations instead of INRs, e.g. state-of-the-art results using image-based (CNNs) or point cloud-based classifiers? Could the authors comment on why the performance is still far from the one that the community has obtained by applying these baselines? This can be clearly observed in CIFAR-10. In addition, apart from the inr2vec baseline, it might be helpful to add the baseline from Dupont et al., ICML’22 (the “functa” representation), where INRs are represented as vectors of latent modulations.
- **NN editing**. In section 3.4., regarding the CIFAR dataset, the differences between the competing methods are not clear qualitatively.  Could the authors comment on this?
- Is there any theoretically-backed advantage of using a combination of equivariant layers followed by an invariant one instead of using only invariant ones (when the ground-truth function is invariant)?


**Limitations:**

- The authors have dedicated a paragraph to the limitations of their method. One thing that might be missing is a clarification (if I understand correctly) that NF layers can only be applied to the weights of neural nets that share the same architecture (fixed number of layers, fixed kernel size for CNNs). Note that L189-190 (“…which contains thousands of CNN weights trained on several datasets with varied hyperparameters”) seems contradictory to this. I think the phrase "varied hyperparameters" should be clarified to avoid confusion.

- Moreover, given that classifying INRs with NF-layers seems to still have room for improvement in order to reach the performance achieved by traditional neural nets that operate on raw representations, it might be useful to discuss the possible reasons for this more thoroughly (perhaps stronger inductive biases, taking into account other properties of the geometry of weight spaces, might be missing?)

- No foreseeable negative societal impact.

---

> ### Author Rebuttal · Authors · 2023-08-10
>
> We appreciate the detailed and insightful review, and agree that indepedently developed frameworks can be useful contributions to the community.
>
> > I found the explanations in section B.4. a bit hard to follow.
>
> We will aim to improve the exposition in Section B.4. We also welcome any feedback on particular aspects that are confusing or could be improved.
>
> > I think that the authors should be more upfront about [Navon et al] and mention this early on [...]
>
> We will update the introduction to more clearly and prominently discuss DWSNets, and our contributions relative to that work. See our top level reply for more detail.
>
> > Have the authors tried learnable [positional encodings] instead? What is the number of added parameters when using learnable positional encodings?
>
> We will update the appendix to be clearer about this: the positional encodings (PE) are sinusoidal--learnable ones are also possible, though in Transformers it was observed that learned vs sinusoidal PE had essentially no impact on performance [1, Table 3 Row E].
>
> For learned PE you would need a separate vector of length $c$ (feature channels) This would add $c(n_0 + n_L)$ parameters.
>
> > How do the authors explain the fact that using positional encodings in NFN_NP performs better than NFN_HNP?
>
> Note the effect is not consistent--the HNP variant performs better in some tasks, such as predicting generalization (Table 2). NFN_NP may perform better on some settings simply because it is more parameter-efficient and easier to train, or other optimization related details. Also, the PE ablation in our top-level reply suggests that differences in performance are largely due to differences in the HNP and NP architectures themselves, rather than PE.
>
> > NFN_PT case. Since this method performs well in the pruning mask prediction experiment, why didn’t the authors also include its performance in the rest of the experiments
>
> This is a good question. We generally didn’t include the pointwise (PT) ablation because, in our early development, PT architectures performed very poorly on all problems _except_ for pruning. Since PT is much more restricted than the full (H)NP layers, poor performance on most settings was expected and we focused our attention on ablations in the setting where it did have signs of life (pruning).
>
> > Pruning: Why did the authors use a generative model (CVAE) for the winning ticket experiment? Why not a deterministic predictor (a 0-1 classifier)? Have the authors tested this? How far are the generated sparsity masks from the ones that IMP yields? Why isn’t the performance of MLP and MLP_aug included in this table?
>
> Even conditioned on a fixed initialization, winning tickets are _not_ deterministic due to the noise of SGD–one can quickly verify this experimentally. Therefore, it makes sense to learn a probabilistic model p(mask | init) rather than a deterministic mapping.
>
> > In section 3.2, would it be possible for completeness to add some baselines that classify images and 3D shapes, using different representations instead of INRs, e.g. state-of-the-art results using image-based (CNNs) or point cloud-based classifiers?
>
> We will add these (non weight-space) baselines in Section 3.2.
>
> > Could the authors comment on why the performance is still far from the one that the community has obtained by applying these baselines? [...] it might be useful to discuss the possible reasons for this more thoroughly (perhaps stronger inductive biases, taking into account other properties of the geometry of weight spaces, might be missing?)
>
> As suggested, we believe that this is primarily due to a difference in inductive biases. For example, it is common to use convolutional networks because of translation symmetry when solving CIFAR in image space. While NFNs are able to leverage weight space symmetries, it does _not_ have symmetry to translations of the CIFAR INRs. How to encode inductive biases related to the underlying natural signal, not just the weights, remains an interesting and challenging direction for future work. We will add this commentary to the paper.
>
> > In addition, apart from the inr2vec baseline, it might be helpful to add the baseline from Dupont et al., ICML’22 (the “functa” representation), where INRs are represented as vectors of latent modulations.
>
> Functa cannot be run on our INR datasets since it requires a special (meta-learning) training process to produce the INRs, while the INRs in our datasets are produced by vanilla training methods. We would also interpret functa as operating in a different setting depending on how much control one has over the process that produces the input weights.
>
> > NN editing. In section 3.4., regarding the CIFAR dataset, the differences between the competing methods are not clear qualitatively. Could the authors comment on this?
>
> For the CIFAR contrast task, it turns out that even the ground truth brightening operation can be relatively subtle to notice visually. Coupled with the fact that none of the methods are perfect at achieving the ground truth operation, this makes the qualitative differences difficult to observe (though some samples are more obvious than others, see the right column of 4). We will investigate to see if there is a better way to display the visual changes.
>
> > NF layers can only be applied to the weights of neural nets that share the same architecture [...] L189-190 (“…which contains thousands of CNN weights trained on several datasets with varied hyperparameters”) seems contradictory to this.
>
> The CNNs in that CNN Zoo share the same architecture but vary in learning rate and other optimization hyperparameters. We will clarify this wording to avoid confusion.
>
> ## References
> [1] Vaswani et al. Attention is all you need.

---

> > ### Comment · Reviewer_jZvx · 2023-08-19
> > **Response to Authors**
> >
> > Dear authors,
> >
> > Thank you for your reply. Most of my concerns have been addressed. I would advise incorporating the explanations you give here in an updated version (e.g. prominently discuss and compare against Navon et al., details about positional encodings including the ablation study, adding non-weight space baselines in 3.2. and discussing the differences). I continue to support the acceptance of the paper and my initial score remains.
> >
> > *Minor*: The only thing I am sceptical about is the absence of comparison to Dupont et al., ICML'22. The authors correctly mention that the INRs in this paper are produced by specialised training, but since with this method one obtains a vector representation of each INR, then this representation can be fed to, e.g., an MLP to solve all the *invariant* tasks tested in this paper. This is not a major concern, but it would be nice to include an experimental comparison (missing from Navon et al. as well). I would also encourage the authors to discuss my last question in an updated version (*“Is there any theoretically-backed advantage of using a combination of equivariant layers followed by an invariant one instead of using only invariant ones (when the ground-truth function is invariant)?”*) – this was not answered in the rebuttal.

---

### Official Review · Reviewer_CEwM · 2023-06-30

**Soundness:** 2 fair
**Presentation:** 3 good
**Contribution:** 2 fair
**Rating:** 3
**Confidence:** 5

**Summary:**

The authors proposed an architecture for processing other networks’ weights and implicit neural representations (INRs). The generalization abilities of this architecture are enhanced and the number of parameters is reduced by leveraging the symmetries of deep networks.

The authors claim the following contributions:
- Proposing a new architecture named Neural Functional Networks (NFN) for processing weights of other networks including INRs. A key building block in NFNs are NFLayers which the authors constrained to be permutation equivariant.
- Extensive experimental part that demonstrates the superiority of NFNs over various baselines.


**Strengths:**

- The paper is dealing with novel and interesting problems of processing networks’ weights and INRs, I believe that this intriguing domain can open the path to more interesting and valuable works for our community.
- The paper is well-written and easy to follow. Specifically, the visualizations make the reading more accessible and straightforward.
- The experimental part is extensive and includes multiple learning setups and datasets.


**Weaknesses:**

- My main concern is about the novelty of this work w.r.t [1]. In [1] the authors also proposed an architecture, coined DWSNets,  for processing weights and INRs. They also provide some theoretical guarantees about the expressive power of their method. It is worth mentioning that [1] is **not a concurrent work** according to the NeurIPS guidelines. Therefore, the authors should explain what is the novelty of the current work over [1].
- Given the previous point, the authors should also include [1] as a baseline in the experimental part and cite it more explicitly in the paper.
- There is a lack of available technical information regarding the process of generating INRs. Specifically, in sections 3.2, 3.3, and 3.4 the authors did not mention the number of samples used for train/val/test. I also wonder how long it took to generate the INRs and how many resources were used. The authors should also include information about the training paradigm of INRs (for example the number of optimization steps) to enable easier reproducibility.
- Although performed in the weight space it will be interesting to see how well NFNs deal with more challenging style editing tasks like inpainting, deblurring, etc. Dilation and contrast editing are straightforward tasks in CV.
- Given that it was the pioneering study demonstrating the permutation equivariance of feed-forward networks, it is advisable for the authors to include a citation to [2] (especially in lines 33-36).
- In lines 155-158 the authors stated they have used positional encoding (PE) to boost NFNs performance (although it breaks the symmetry), what is the performance gain presented by using PE? I did not see such an ablation (although I might have missed it).
----------
Citations:

[1] Equivariant Architectures for Learning in Deep Weight Spaces, Navon et al.

[2] On the algebraic structure of feedforward network weight spaces, JHecht-Nielsen et al.


**Questions:**

- Can NFNs handle heterogeneous networks in a single dataset? For example networks with varying input dimensions and hidden features.
- What is the computational complexity of NFN / NFLayers?


**Limitations:**

A limitation section is included.

---

> ### Author Rebuttal · Authors · 2023-08-10
>
> Thank you for your suggestions and questions--we aim to clarify our contributions and strengthen the experiments with the suggested baselines.
>
> > explain what is the novelty of the current work over [DWS].
>
> We agree that DWSNets are a very relevant recent work with significant overlaps and notable differences. We will update the introduction to more clearly and prominently discuss DWSNets, and our contributions relative to that work--see the top-level reply for details.
>
> We would like to emphasize some contributions of  our work relative relative to DWS:
>
> 1. We additionally focus on the NP setting, which leads to layers that are more scalable and much more parameter-efficient (see Table 1) than HNP/DWS layers while maintaining good performance in most of our experiments. NP layers are also easier to visualize, understand, and implement, as evidenced by Figure 2 and the pseudocode in Appendix A.
> 1. We extend both HNP/DWS and NP layers to handle CNN weights, in addition to MLPs. This extension is not trivial as CNN filters have additional dimensions that are not permutable, so the symmetries of CNN weights and MLP weights are not the same.
>
> With regards to technical foundations, as mentioned by Reviewer jZvx, independently developed approaches for deriving equivariant weight-processing layers can be a useful contribution to the community. DWSNets and our paper also contain many complementary experiments and benchmarks to demonstrate the applicability of these architectures.
>
> > Given the previous point, the authors should also include [DWS] as a baseline in the experimental part
>
> We have run DWSNets on our 2D-INR classification datasets and present the results in our top-level reply. DWSNets perform somewhat worse than both NFN variants across the board, even when controlling the number of parameters and after sweeping learning rates. We will also include results for this baseline on the other applicable benchmarks in our revised paper.
>
> > The authors should also include information about the training paradigm of INRs (for example the number of optimization steps) to enable easier reproducibility.
>
> Appendix D.3 contains additional information about the INR datasets, including the train/val/test sizes and the optimization process for generating each INR. We will add more explicit references to this information in Sections 3.2-3.4 of the main paper. Producing each INR takes ~168 seconds with 2 CPU cores (no GPU required), and can be done in parallel over multiple machines/cores. For the easiest reproducibility, we will also publicly release the INR datasets after the anonymous period.
>
> > Although performed in the weight space it will be interesting to see how well NFNs deal with more challenging style editing tasks like inpainting, deblurring, etc.
>
> As you note, although the style editing tasks are easy in image space the primary challenge is to be able to accomplish them in weight space, which is much more difficult. We agree that more challenging image editing tasks like inpainting and deblurring are a worthwhile goal, but ultimately become more of a test of geometric inductive biases (which convolutional networks have) rather than inductive biases related to weight space symmetries, which is the focus of this work.
>
> > It is advisable for the authors to include a citation to [2] (especially in lines 33-36).
>
> Our paper already cites [2] on Line 31, but we will also add a citation in lines 33-36.
>
> > used positional encoding (PE) to boost NFNs performance (although it breaks the symmetry)
>
> The input/output positional encoding (PE) for the NP variant of NFNs _only breaks symmetry to permutations of the input and output neurons of the weight space_, but preserves symmetry to hidden layer permutations. We apologize for the confusing wording and will update the writing.
>
> > What is the performance gain presented by using PE?
>
> We performed an ablation to answer this question--see the top-level reply for details. Results show that PE actually adds a very small (sometimes negligible) boost to NFN_NP performance. This suggests that the NFN_NP architecture by itself can solve many weight space tasks without needing to break permutation symmetry of the input and output neurons.
>
> > Can NFNs handle heterogeneous networks in a single dataset?
>
> In principle, parameter sharing makes our NP-equivariant NF-layer agnostic to the widths of the input weights, i.e. the number of neurons at each layer can vary. Similarly, the HNP-equivariant NF-layer is agnostic to the number of neurons at the hidden layers, but the input and output dimensions must stay fixed. The depth (number of layers) of the input weights must be fixed in both cases.
>
> In practice, handling heterogenous networks is challenging in modern ML frameworks since varying input sizes makes batch computation difficult on GPUs, so in our experiments the input networks have a fixed size. In principle, padding could mitigate this problem, though it may use GPU memory inefficiently.
>
> > What is the computational complexity of NFN / NFLayers?
>
> Suppose the input weights have constant width $n=n_0=\cdots=n_L$, so that there are $Ln^2$ input weights in total. A naive linear layer operating on these weights would require $L^2n^4$ operations.
>
> For the NP case, consider implementing Eq 3 without parallelization, by:
> 1. First calculate all quantities of the form $W\_{\star,\star}^{(i)}$, $W\_{j,\star}^{(i)}$, $W\_{\star,k}^{(i)}$. These require $O(Ln^2)$ operations.
> 1. Calculate each term in Eq (3) separately, for each output index $(i,j,k)$. The first term requires $L^2$ operations, terms 2-5 require $Ln$ operations, and the final term requires $Ln^2$ operations.
> 1. Add the terms to produce the final result for each $(i,j,k)$. This requires $O(Ln^2)$ operations.
>
> So the layer can be implemented on $O(L^2 + Ln^2)$ operations. We will include this and the HNP result in the revised paper.

---

### Official Review · Reviewer_VWST · 2023-07-06

**Soundness:** 4 excellent
**Presentation:** 4 excellent
**Contribution:** 3 good
**Rating:** 6
**Confidence:** 4

**Summary:**

This paper considers the design of architectures whose inputs are the parameters of neural networks. They propose an equivariant weight-sharing scheme based on the permutational symmetries of neural networks: one can permute at least the internal neurons (the “HNP” case), and sometimes the input/output neurons as well (the “NP” case), of a network without changing the function represented by the end-to-end network. The parameter-sharing of the “NP” case is more advantageous but less often applicable, since the underlying problem must have additional symmetry structure to warrant input/output neuron permutational symmetries. To remedy this, the authors propose coupling the stronger “NP” case with positional encodings. They extend their framework to take as input convolutional layers, and test their proposed architecture on tasks including image classification from implicit neural representations, predicting sparsity masks, and weight-space editing. They compare to the use of MLPs with and without permutational augmentations, as well as problem-specific baselines, and obtain promising experimental results.

**Strengths:**

Originality: Outside of concurrent work, the idea of using permutational symmetries to train directly on weight and bias inputs is original and makes sense. This permutation subgroup is distinct from other groups that have been studied in the past (such as $S_n$ and $S_n \times S_n$). Even beyond concurrent work, the idea of using positional encodings in conjunction with input/output permutational symmetries is a creative contribution.

Quality: The experiments are varied and test against a reasonable set of baselines.

Clarity: The paper is generally quite clearly written and explained.

Significance: This paper enables the application of weight-space symmetry techniques to CNNs. The problem of learning with neural network weight inputs is topical, and such methods are likely to enjoy practical usage.


**Weaknesses:**

1. The primary weakness of this paper is its novelty in relation to other work by Navon et al (ICML 2023), which also articulates an architecture equivariant to the permutational symmetries of weight space and provides more thorough results on its universality. The authors state this work was concurrent (which I will take at face value). If its novelty is judged in relation to this work, then the main contributions are the NP setting with positional encodings, the extension to CNNs, and the distinct set of experiments, which are extensions of the key idea. This very relevant work is currently only mentioned near the end of the paper, but I would think it is important to have a more in-depth discussion of how the two fit together and the novelty of this work, and for this discussion to appear earlier in the paper as part of its framing.
2. The paper only considers the permutational symmetries of neural networks, which are indeed perhaps the most general symmetries if one does not specify a certain nonlinearity. However, as noted in Godfrey et al 2022, a given nonlinearity may enjoy additional symmetries — for instance, ReLU enjoys a symmetry to positive scaling. This is not discussed in the paper, but would have been a more substantial contribution relative to Navon et al (2023). For instance, one straightforward way of incorporating this scaling symmetry could be to pick a positive scale based on the norm of the parameters of the first layer, and then use this scale to normalize the rest of the input weights and biases.

**Questions:**

1. Is there some advantage to classifying implicit neural representations, instead of the pixels directly? It would be helpful to motivate these problems more in the main body of the text.
2. Can the authors comment on the expressivity of their proposed idea of using input encodings to adapt the NP setting to problems without input/output permutational symmetries? Is this universal, for example, or is there some loss of information from adding the positional encodings? (On a related note, how exactly are the positional encodings incorporated — are they literally added, or are they appended?)
3. Could you be more specific in designing networks for a particular nonlinearity with different symmetries (e.g. scaling for ReLU)? See e.g. “On the Symmetries of Deep Learning Models and their Internal Representations” (Godfrey et al 2022), as well as other references in Navon et al 2023.
4. How large are the models input to the neural functional network? Will all weight-space methods will fail for truly large weight space inputs, i.e. millions of parameters, even if you have very few learnable parameters thanks to symmetry?

**Limitations:**

There is not potential negative societal impact. The authors are upfront that the HNP case is less scalable than the NP case. One limitation not discussed is that the architecture described only takes into account permutational symmetries, and not other nonlinearity-dependent symmetries.

---

> ### Author Rebuttal · Authors · 2023-08-10
>
> Thank you for your insightful review, and for highlighting aspects of our contribution such as the NP setting and application to CNNs.
>
> > it is important to have a more in-depth discussion of how [DWS (Navon et al) and NFN] fit together and the novelty of this work, and for this discussion to appear earlier in the paper as part of its framing.
>
> We will update the introduction to more clearly and prominently discuss DWS, and our contributions relative to that work–see the top-level reply for more details.
>
> > Is there some advantage to classifying implicit neural representations, instead of the pixels directly? It would be helpful to motivate these problems more in the main body of the text.
>
> The results in Section 3.2 represent first steps towards better methods that operate directly on INRs, which we believe will eventually have multiple advantages over operating on discrete representations of data (pixels, point clouds, voxel grids, etc…):
>
> 1. INRs are continuous and decouple the memory cost of the representation from the actual spatial resolution. This is important as we eventually move towards more complex and high resolution 3D signals such as entire 3D scenes [1], where for example working with gridded representations becomes less tractable. We believe the 3D shape SDF experiments are a promising first result for that direction (Table 4).
> 1. Working with INRs directly opens the possibility for a single method that can elegantly classify different types of signals of different sizes and resolutions, or even different dimensions. Whereas, for example, changing the image resolution or size in pixel space can pose a problem for CNNs.
>
> We will add this motivation to Section 3.2.
>
> > Can the authors comment on the expressivity of their proposed idea of using input encodings to adapt the NP setting to problems without input/output permutational symmetries? Is this universal, for example, or is there some loss of information from adding the positional encodings?
>
> This is an interesting question–we don’t develop any such theoretical results for the NP architectures with positional encodings in this work. One could likely develop a universality result through a process analogous to Thm 3 in [2], which shows that positional encodings remove the permutation equivariance constraint and allow Transformers to approximate any function under certain conditions.
>
> > On a related note, how exactly are the positional encodings incorporated — are they literally added, or are they appended?)
>
> In practice, we implement the positional encoding by concatenation (in the channel dimension), though adding the encoding should have a similar effect. For matrices other than the input/output, we simply append zeros to keep the channel dimension consistent.
>
> > The paper only considers the permutational symmetries of neural networks [...] One straightforward way of incorporating [ReLU] scaling symmetry could be to pick a positive scale based on the norm of the parameters of the first layer, and then use this scale to normalize the rest of the input weights and biases.
>
> It is true that NFNs only consider permutation symmetries, while (depending on the activation) scaling symmetries may also exist in the weight space. We will discuss this limitation in the paper.
>
> Since NFNs are typically dealing with weights produced by an optimization process like SGD, some existing literature suggest that accounting for scale symmetry may be unnecessary in practice; to quote [3]
>
> > SGD’s implicit regularization balances weight norms and, therefore, scale invariance does not seem to play an important role in understanding symmetries of solutions found by SGD.
>
> This could explain why we see decent performance on some tasks already. The suggested idea seems very interesting, though we likely do not have time to try it out in the limited discussion timeframe.
>
> > How large are the models input to the neural functional network? Will all weight-space methods fail for truly large weight space inputs, i.e. millions of parameters, even if you have very few learnable parameters thanks to symmetry?
>
> The size of the inputs to the NFN depends on the task: for example in predicting winning tickets we have 3 layers and 128 neurons, while INR classification deals with 3-layer INRs having 32 hidden neurons each (see Appendix D for full details). It is possible that much larger weight spaces will be challenging to learn in for any NF-type architecture, though such weight spaces also pose practical problems due to increased memory and compute usage.
>
> ## References
> [1] Mildenhall et al. Representing Scenes as Neural Radiance Fields for View Synthesis.
>
> [2] Yun et al. Are Transformers universal approximators of sequence-to-sequence functions?
>
> [3] Entezari et al. The Role of Permutation Invariance in Linear Mode Connectivity of Neural Networks.

---

> > ### Comment · Reviewer_VWST · 2023-08-20
> > **Thank you for the response**
> >
> > Thank you to the authors for their response. They clarified the use of operating on INR representations, made a valid point about why scale symmetry may be superfluous, and generally answered my other questions. My only remaining concern retains to the novelty of this work relative to Navon et al. I think the experimental comparison and added paragraph will help on this front, but because the contributions relative to Navon et al are still fairly minor, I retain my original rating of weak accept.

---

### Author Rebuttal · Authors · 2023-08-10

We thank the reviewers for their detailed and thoughtful comments and questions. Reviewer suggestions have helped us improve the writing and pointed us towards additional experiments that significantly strengthen the paper. A brief summary of changes and new experiments:

* We will update the introduction to more clearly and prominently discuss DWSNets, and our contributions relative to that work.
* We have run DWSNets on our benchmarks in order to provide a direct comparison.
* We performed ablation experiments on the effect of positional encoding (PE) on the NFN_NP architecture variant.

Additional changes and information are provided in the reviewer-specific replies.

## DWSNet discussion

In order to more clearly discuss DWSNets as a relevant work with significant overlap and notable differences, we will put the following discussion to the introduction:

> The recent work of Navon et al. [45] recognized the potential of leveraging weight space symmetries to build equivariant architectures on deep weight spaces; they characterize a weight-space layer which is mathematically equivalent to the equivariant NF-Layer we develop for the HNP setting. Their work additionally studies interesting universality properties of the resulting equivariant architectures, and demonstrates strong empirical results for a suite of tasks that require processing the weights of MLPs.


> Our independently developed framework additionally focuses on the NP setting, where making stronger symmetry assumptions enables us to develop equivariant layers with improved parameter efficiency and practical scalability. This work also extends both NFN variants to process convolutional neural networks (CNNs) as input, leading to applications such as predicting the generalization of CNN classifiers (Section 3.1).


## DWSNet Comparison

Here is an initial empirical evaluation of DWS on our INR classification benchmarks. We trained DWSNets both at width 32 (as in [1]) and width 512 (as in ours), with our data augmentation of creating multiple INR copies per image. We also tried [1]'s data augmentation scheme, but found that it tended to hurt performance. For each dataset and channel size we swept learning rates in $[1e-3, 5e-3, 1e-4, 5e-4]$. Initial results show somewhat lower test accuracies than NFNs, with comparable performance on CIFAR10. Although the DWS layers match our HNP variant in theory, real world performance depends heavily on many architectural hyperparameters, just as different CNN architectures can achieve different performance.

| Test accuracy  | MNIST | FashionMNIST | CIFAR |
|----------------|-------|--------------|-------|
| DWS (32 channel)| 74.7 | 67.5         | 42.3  |
| DWS (512 channel)| 61.6 | 62.0        | 42.9  |
| NFN_NP (512 channel)| 92.9 | 75.6      | 46.6  |
| NFN_HNP (512 channel)| 92.5 | 72.7     | 44.1  |

And here is the number of parameters for each method, for reference:
| No. params     | MNIST | FashionMNIST | CIFAR |
|----------------|-------|--------------|-------|
| DWS (32 channel)| 0.6M | 0.6M         | 1M  |
| DWS (512 channel)| 71M | 71M         | 134M  |
| NFN_NP (512 channel)| 45M | 45M       | 47M   |
| NFN_HNP (512 channel)| 69M | 69M      | 135M  |

## Positional Encoding (PE) ablation

We performed the PE ablation on both 2D INR classification and style editing tasks. The results show that PE actually adds a very small (sometimes negligible) boost to NFN_NP performance, though it never hurts. Since NFN_NP often performs as well as or better than NFN_HNP, this indicates that even the base NP variant can solve many weight space tasks without needing to break that symmetry.

**Table 3b**: Ablating the PE on the INR classification. Higher is better.
|             | NFN_NP         | NFN_NP (no PE) |
|-------------|----------------|----------------|
| CIFAR-10    | 46.6 ± 0.072   | 46.5 ± 0.160   |
| MNIST       | 92.9 ± 0.218   | 92.9 ± 0.077   |
| FashionMNIST| 75.6 ± 1.07    | 73.4 ± 0.701   |

**Table 6b**: Ablating the PE on style editing tasks. Lower is better.
|                | NFN_NP       | NFN_NP (no PE) |
|----------------|--------------|----------------|
| Contrast (CIFAR-10) | 0.020 ± 0.000 | 0.020 ± 0.000 |
| Dilate (MNIST) | 0.068 ± 0.000 | 0.070 ± 0.001 |

## References
1. Navon et al. Equivariant Architectures for Learning in Deep Weight Spaces.

---

### Decision · Program_Chairs · 2023-09-21

**Decision:**

Accept (poster)

**Comment:**

The paper proposes an equivariant architecture for processing the weights of neural networks. Reviewers were generally positive with regard to the idea and the paper. A concern was raised regarding the limited contribution of this paper with respect to a recent ICML 2023 paper by Navon et al.
The authors submitted a rebuttal, acknowledged the similarity to this previous work, mentioned the differences, and promised to prominently discuss it in the revision.
In the discussion, most reviewers agreed that although these two papers have a lot in common, the current paper has some new contributions like the extension to CNN, the NP setup (permutation equivariance on input and output dimensions), and additional interesting experiments.
The AC has read the current paper and the paper by Navon et al. thoroughly. It is the AC's opinion that the extension to convents is a marginal contribution. The NP setting with the positional encoding is the more important contribution as it allows scaling the networks in an elegant way. In the discussion, the AC and reviewers agreed to accept the paper. The authors are asked to revise their paper in accordance with the comments above and their rebuttal.